# Coconut Milk-Derived Bioactive Peptides as Multifunctional Agents Against Hyperglycemia, Oxidative Stress, and Glycation: An Integrated Experimental and Computational Study

**DOI:** 10.3390/ijms27010360

**Published:** 2025-12-29

**Authors:** Akshaya Simha Naganarasimha, Shashank M. Patil, Ramith Ramu, Maciej Przybyłek, Piotr Bełdowski, Olga Małolepsza, Sławomir Bujanowski, Mudassar Shahid

**Affiliations:** 1Department of Biotechnology and Bioinformatics, JSS Academy of Higher Education and Research, Mysuru 570015, Karnataka, India; akshayasimhan@jssuni.edu.in (A.S.N.); shashankmpatil@jssuni.edu.in (S.M.P.); 2Department of Physical Chemistry, Pharmacy Faculty, Collegium Medicum of Bydgoszcz, Nicolaus Copernicus University in Toruń, 85096 Bydgoszcz, Poland; m.przybylek@cm.umk.pl; 3Institute of Computer Science, Kazimierz Wielki University in Bydgoszcz, 85-064 Bydgoszcz, Poland; piotrbel@ukw.edu.pl (P.B.); olga.malolepsza@ukw.edu.pl (O.M.); 4Faculty of Telecommunications, Computer Science and Electrical Engineering, Bydgoszcz University of Science and Technology, Al. Prof. S. Kaliskiego 7, 85-796 Bydgoszcz, Poland; slawomir.bujnowski@pbs.edu.pl; 5Department of Pharmaceutics, College of Pharmacy, King Saud University, Riyadh 11451, Saudi Arabia; mahmad1@ksu.edu.sa

**Keywords:** coconut milk, bioactive peptides, anti-hyperglycemic, anti-glycation, antioxidant, in silico, molecular dynamics simulations, docking, bioactivity prediction

## Abstract

Type 2 diabetes mellitus (T2DM) is characterised by chronic hyperglycaemia and accumulation of advanced glycation end products (AGEs), driving interest in food-derived peptides as safer multifunctional modulators. Coconut milk is a promising source, but its anti-hyperglycaemic and anti-glycation potential remains largely unexplored. Here, proteins from coconut cream, skimmed and insoluble fractions of coconut milk were enzymatically hydrolysed, and the resulting peptides were profiled by nano-ESI-Orbitrap-LC-MS/MS. One hundred and fourteen peptides were identified and screened in silico against α-glucosidase, α-amylase, aldose reductase and the receptor for AGEs (RAGE). Two peptides, MQIFVK and ADVFNPR, showed the most favourable docking scores and physicochemical properties. However, ADVFNPR inhibited all 3 diabetic targets & RAGE. Molecular dynamics analysis showed that both peptides bind stably to the diabetic targets. Both peptides were synthesised and evaluated in vitro. ADVFNPR significantly inhibited α-glucosidase, α-amylase and aldose reductase with lower IC_50_ values and displayed competitive inhibition kinetics. It also scavenged methylglyoxal, 2,2-diphenyl-1-picrylhydrazyl (DPPH), 2,2′-azino-bis(3-ethylbenzothiazoline-6-sulfonic acid (ABTS) and superoxide radicals at low EC_50_ values, and showed low hemolytic activity in human erythrocytes. These findings indicate that coconut milk contains multifunctional peptides with anti-hyperglycaemic, anti-glycation and antioxidant activities that may be further developed as food-derived adjuncts for managing T2DM and glycation-related complications.

## 1. Introduction

Type 2 Diabetes mellitus (T2DM) is a long-term metabolic disorder marked by ongoing high blood sugar levels after meals, mainly caused by rapid breakdown of carbohydrates and poor glucose use. Enzymes like α-amylase and α-glucosidase break down dietary starch into glucose that can be absorbed. Still, their high activity can cause significant postprandial blood sugar spikes in type 2 diabetes. Besides abnormal glucose absorption, sustained high blood sugar increases glucose flux into cells through alternative pathways, such as the polyol pathway, in which aldose reductase converts glucose to sorbitol, leading to osmotic stress and microvascular damage. Extended hyperglycemia also fosters the formation of reactive compounds, such as methylglyoxal, which promote non-enzymatic glycation and the buildup of advanced glycation end products (AGEs), major contributors to diabetic complications [1,2].

The accumulation of AGEs leads to dysfunction of β-cells, insulin resistance, endothelial damage, and tissue inflammation, primarily through activation of the receptor for advanced glycation end products (RAGE). The interaction between AGEs and RAGE initiates downstream signalling pathways, including mitogen-activated protein kinase (MAPK), Janus-Kinase/Signal Transducers and Activators of Transcription (JAK/STAT), and nuclear factor-kappa B (NF-κB), that result in ongoing oxidative stress, inflammatory responses, and gradual damage to renal and vascular tissues [3,4,5]. Together, these pathways emphasise the complex nature of diabetes progression, suggesting that practical treatment approaches should simultaneously address postprandial hyperglycemia, cellular glucose flux, and tissue injury caused by glycation. Therefore, targeting α-amylase, α-glucosidase, and aldose reductase, in addition to reducing AGE formation and RAGE signalling, constitutes a sensible multi-target strategy for alleviating diabetes and its related complications.

Due to the long-term adverse effects associated with traditional synthetic antidiabetic medications, there is a rising interest in natural, food-derived bioactive compounds that can inhibit multiple targets. Among these, bioactive peptides derived from dietary proteins have attracted significant attention due to their safety, high bioavailability, and ability to modulate key metabolic enzymes and oxidative processes. Recent research has identified potent α-glucosidase and α-amylase inhibitory peptides from various food sources, including fermented spent coffee grounds, lupin protein hydrolysates, and oat proteins [6,7,8]. Several of these peptides show strong enzyme binding, low IC_50_ values, and promising in silico pharmacokinetic profiles. These results underscore the increasing recognition of food-derived peptides as viable alternatives for controlling hyperglycemia and its related complications.

Despite advances, plant-based milk systems remain relatively understudied sources of antidiabetic bioactive peptides. Coconut milk (CM), a rich emulsion derived from the grated endosperm of mature coconuts, is a nutritionally valuable yet little-explored protein source. It contains a distinctive protein fraction enriched in essential and semi-essential amino acids, including arginine, lysine, and glutamic acid, which contribute to enzyme inhibition, antioxidant activity, and dicarbonyl scavenging. Coconut milk stands out for having one of the highest arginine contents among plant-based milks, a trait associated with better glucose metabolism and bioactivity [9,10,11]. Nevertheless, unlike its lipid component, coconut milk proteins and peptides have been insufficiently studied, particularly regarding their potential antidiabetic and anti-glycation effects.

Due to the favourable amino acid profile of coconut milk proteins and the proven effectiveness of food-derived peptides in inhibiting carbohydrate-digesting enzymes and glycation processes, coconut milk is a promising yet underexplored source of multifunctional bioactive peptides. No comprehensive studies have yet evaluated the combined inhibitory activities of coconut milk-derived peptides against α-amylase, α-glucosidase, and aldose reductase, as well as their antiglycation potential. This study aims to identify, characterise, and assess these bioactive peptides through combined in silico and in vitro methods, focusing on their anti-hyperglycemic, antioxidant, and anti-glycation effects. Ultimately, this research seeks to position coconut milk as a novel plant-based source of multifunctional peptides and to lay the scientific groundwork for the development of food-derived therapeutic options for diabetes management.

## 2. Results and Discussion

### 2.1. Extraction, Estimation, and Profiling of Coconut Milk Proteins

A total of 4 coconuts were purchased from the local farm, yielding 527 mL of opaque, milky-white liquid, commonly known as coconut milk. This coconut milk was further centrifuged to extract the proteins. After centrifugation, three fractions were obtained: 101.95 g of coconut cream (top fraction), 450 mL of skimmed coconut milk (middle fraction), and 10.48 g of an insoluble pellet (bottom fraction). All three sources were used for protein extraction according to the protocol outlined in the methodology section. The extracted proteins were estimated using the Bicinchoninic acid (BCA) method. The results of protein estimation are depicted in Table 1. The concentration of protein present in coconut milk is much less compared to rice milk (0.28–2%) and soya milk (3.82–3.98%). The lower protein content could also be attributed to techniques such as dialysis and dilutions used during protein extraction. However, protein concentration varies depending on the type of coconut used for coconut milk extraction [12].

Sodium Dodecyl Sulphate-Polyacrylamide Gel Electrophoresis (SDS-PAGE) was performed to profile the proteins in coconut milk. The results are represented in Figure 1. The SDS-PAGE results suggest that the proteins in coconut milk have molecular weights greater than 10 kDa. The results are in line with the previous study by Kwon et al. [13], which identified five different proteins: albumins, globulins, prolamines, glutelins-1, and glutelins-2. According to Senphan and Benjakul [14], the primary protein in coconut milk has a molecular weight of 56 kDa. Additionally, proteins with molecular weights of 31, 25, 19, 17, and 16 kDa were identified in coconut milk. Proteins weighing less than 19 kDa were also detected within it. The findings of this study align with current research, which observed a thick, bright band between 50 and 60 kDa, likely corresponding to Cocosin [15]. The predominant protein in coconut (65%) is 11S globulin, known as Cocosin [16]. This protein is a hexamer composed of 55 kDa subunits, each of which contains an acidic (32–34 kDa) and a basic (22–24 kDa) polypeptide linked by a disulfide bridge [15,16]. Cocosin is thought to be more crucial in maintaining the stability of coconut milk compared to albumin or the 7S globulin fraction [15].

### 2.2. Identification of Peptides by Nano-ESI-Orbitrap-LC-MS/MS

The coconut milk protein hydrolysates obtained after the simulated gastrointestinal digestion were concentrated using a 3 kDa ultrafiltration membrane and desalted before analysis by nano-Electrospray Ionization Orbitrap-Liquid Chromatography-Mass spectrometry/Mass spectrometry (nano-ESI-Orbitrap-LC-MS/MS). Peptide identification was performed by comparison against the *Cocos nucifera* proteome, resulting in the confident assignment of 114 unique peptide sequences. Representative high-resolution chromatograms are shown in Figure 2. The biological significance depends on their physicochemical and structural properties. Most peptides range from 6 to 15 amino acids in length and have molecular weights below 2.5 kDa. This size range is generally considered advantageous for stability in the gastrointestinal tract, facilitating intestinal transport and interaction with metabolic enzymes, thereby supporting potential bioavailability after oral intake. Furthermore, medium-length peptides possess sufficient structural complexity to bind effectively to enzyme active or allosteric sites, unlike very short di- or tripeptides, which tend to have limited binding capacity. In the current study, the identified peptide sequences were enriched in hydrophobic and aromatic residues, including valine and phenylalanine, as well as in basic residues such as arginine and lysine. These amino acid features are commonly associated with bioactive food-derived peptides and are known to promote hydrophobic interactions, π–π stacking, hydrogen bonding, and electrostatic contacts within protein targets [17,18]. Such interactions are particularly relevant for enzymes involved in postprandial glucose regulation and polyol pathway flux, including α-glucosidase, α-amylase, and aldose reductase. In addition, the presence of basic residues may contribute to the scavenging of reactive carbonyl species, thereby supporting potential antiglycation activity [19,20].

Compared with previous peptidomic studies on other plant matrices, variations in peptide numbers and profiles largely reflect differences in protein composition, digestion strategy, and analytical depth. While fewer peptides have been reported for oat and egg-derived hydrolysates analysed under more limited digestion or mass spectrometric conditions, more extensive multi-enzyme hydrolysis combined with high-resolution nano-LC–MS/MS has been shown to yield substantially larger peptide datasets in complex protein systems [21,22]. In this context, the identification of 114 peptides from coconut milk proteins provides moderate-to-high coverage, consistent with a robust analytical workflow, while maintaining a peptide size range relevant to biological function.

Notably, the peptides identified in this study were produced via simulated gastrointestinal digestion with pepsin and pancreatic enzymes, which offers greater physiological relevance than single-enzyme hydrolysis methods. This approach is likely to mimic better the peptides that could be available for absorption and interaction with metabolic targets within the body. While discovering more peptides may not indicate increased biological activity, a structurally diverse set of peptides increases the likelihood of finding sequences with desirable binding and multifunctional characteristics.

Overall, the peptidomic profile of coconut milk protein hydrolysates reveals a diverse collection of low-molecular-weight, medium-length peptides enriched in amino acids associated with enzyme inhibition, antioxidant activity, and antiglycation potential. These characteristics provide a mechanistic rationale for the subsequent in silico docking analyses and in vitro evaluations, positioning coconut milk proteins as a promising source of multifunctional bioactive peptides relevant to glycation-associated metabolic dysfunction.

### 2.3. In Silico Screening and Identification of Lead Peptides

The nano-ESI-Orbitrap-LC-MS/MS analysis indicated the presence of 114 peptide sequences, of which 96 unique peptides were retained after removal of duplicates. These 96 peptides were characterised in silico for PeptideRanker bioactivity score, sequence length (6–15 amino acids), molecular formula, molecular weight (target < 2.5 kDa), net charge, isoelectric point (pI), numbers of positively and negatively charged residues, instability index, extinction coefficient, hydrophobicity, water solubility, allergenicity and toxicity, using PeptideRanker, ProtParam, PepCalc, AlgPred and ToxinPred. The complete in silico characterisation of coconut milk peptides is summarised in Appendix A.

Among the 96 peptides, 51 were predicted to be stable, with an instability index below 40. Seventeen of these stable peptides had 10 or fewer amino acid residues, whereas the remaining 35 were longer but within 20 residues. Forty-three peptides had a molecular weight above 1000 g/mol, and eight had a molecular weight below 1000 g/mol. With respect to water solubility, 44 peptides were predicted to be soluble and seven to have poor solubility. All 96 peptides were classified as non-toxic, while 29 were predicted as probable non-allergens and 22 as potential allergens. Within the group of 51 stable peptides, 17 had a PeptideRanker score above 0.5 and were therefore considered bioactive candidates. These findings are consistent with previous in silico screenings of food-derived peptides, such as oat kernel peptides evaluated by Darewicz et al. [23] using a comparable computational workflow.

The 17 stable bioactive peptides were subsequently docked against three targets relevant to hyperglycaemia: α-glucosidase, α-amylase, aldose reductase, and one target related to glycation, RAGE. Among these, two sequences, peptide 1 (MQIFVK), inhibited all three diabetes related targets; however, it did not inhibit RAGE. Whereas peptide 2 (ADVFNPR) inhibited all three enzymes and also RAGE. The detailed docking results are summarised in Table 2. The docking scores obtained in the present study compare favourably with those reported by Wang et al. [24], who docked peptides derived from oat milk globulins against α-glucosidase and observed docking scores ranging from −7.1 to −9.2 kcal/mol. In contrast, the coconut milk-derived peptides identified here reached docking scores around −9.3 kcal/mol or lower (i.e., more negative), indicating a higher predicted binding affinity than oat milk globulin-derived peptides. This enhanced affinity is likely related to the presence of hydrophobic and aromatic residues such as Val, Phe and Pro, which promote π–π stacking and hydrogen bonding within catalytic pockets [25].

Recent work further supports the importance of these structural features for inhibitory activity. Fadimu et al. [17] reported that peptides from lupin protein hydrolysates displayed docking scores between approximately −6 and −9 kcal/mol against α-glucosidase and α-amylase, forming hydrogen bonds and hydrophobic interactions with key residues such as Trp59, Tyr62, His299 and Asp197. In this context, coconut milk peptides achieving docking scores at or below −9.3 kcal/mol fall at the more favourable end of the range reported for food-derived peptide inhibitors under comparable in silico conditions.

Beyond α-glucosidase and α-amylase, the coconut peptides also showed significant interactions with aldose reductase (AR), a central enzyme in the polyol pathway associated with hyperglycaemia-induced oxidative stress. Oyebamiji et al. [26] developed peptide inhibitors of AR with the most favourable binding energies around −8.1 kcal/mol, comparable to or better than traditional drugs such as metformin. The docking scores obtained here for coconut-derived peptides against AR are numerically more favourable than −8.1 kcal/mol, suggesting a stronger predicted ability to inhibit AR and potentially reduce sorbitol accumulation and cellular damage in diabetic conditions. Moreover, the observation that peptide 2 also interacts with RAGE is both novel and relevant, as reports on peptide-based RAGE inhibitors remain scarce. Given the central role of RAGE in glycation-driven oxidative stress and long-term diabetic complications, the capacity of coconut peptides to target both carbohydrate-digesting enzymes and RAGE underscores their multifunctional potential, in line with current strategies that aim to modulate multiple metabolic and signalling pathways simultaneously [27,28].

From a structure–activity perspective, the predominance of hydrophobic, proline-rich and aromatic amino acid residues in these sequences likely contributes to their stability and binding specificity. Such residues support π–π stacking, van der Waals interactions and hydrogen bonding with catalytic and substrate-recognition residues in the target enzymes, which may account for their superior docking performance relative to several previously reported food-derived peptides. These observations are consistent with recent structure–function analyses of antidiabetic peptides [17,18]. Compared with other plant-based milks, such as soy and oat, which mainly yield single-target carbohydrase inhibitors [29], the dual interaction of coconut peptides with aldose reductase and RAGE, in addition to α-glucosidase and α-amylase, highlights their potential to attenuate both hyperglycaemia and glycation-associated oxidative stress.

Representative 2D interaction diagrams for peptide 1 and peptide 2 with α-glucosidase, α-amylase, aldose reductase and RAGE are provided in Appendix A, and the key amino-acid contacts for the peptides and control drugs are summarised in Appendix A.

### 2.4. Molecular Dynamics Analysis of Peptide and Reference Drug Complexes

Molecular dynamics (MD) simulations were used to characterise the dynamic behaviour of peptides 1 and 2 in complexes with α-glucosidase, α-amylase, aldose reductase and RAGE, and to compare their stability and binding persistence with those of the reference inhibitors acarbose, quercetin and papaverine. This comparison provides a basis for relating the dynamic features of the complexes to their inhibitory activity.

For peptide 1, 100 ns molecular dynamics simulations were performed on its complexes with the three diabetic targets: α-glucosidase, α-amylase, and aldose reductase. The time evolution of protein backbone root mean square deviation (RMSD) and root mean square fluctuations (RMSF), together with ligand RMSD, RMSF, radius of gyration and surface descriptors such as molecular surface area (MolSA), solvent accessible surface area (SASA), and polar surface area (PSA), is summarised in Figure 3. The corresponding trajectories for the reference inhibitors acarbose and quercetin, as well as for papaverine bound to RAGE, are shown in Appendix A and serve as a benchmark for the comparative remarks below.

The peptide 1-α-glucosidase complex’s RMSD (Figure 3A) rose from ~1.0 to 1.6 Å initially, then fluctuated between 1.6–2.7 Å with stability around 1.8–2.4 Å, indicating a stable folded state. Ligand RMSD stayed low (0.6–1.2 Å) for 38 ns before sharply increasing to ~7 Å at 38–40 ns, suggesting conformational change or partial unbinding. RMSF (Figure 3D) revealed flexibility near residues ~160, ~250, and ~430 (RMSF > 2.0 Å), while other regions were rigid (RMSF < 1.0 Å). Essential residues showed mixed flexibility, implying a balance of stability and mobility. Ligand RMSF (Figure 3G) indicated high flexibility (above 8 Å) at atoms 9–11 and a rigid core (1–2 Å) at 20–30. The combined analysis (Figure 3J) showed RMSD rising from ~1.5 to 3.0 Å, then stabilising, with the radius of gyration between 7.2 and 7.8 Å. About 1–3 hydrogen bonds formed after 35 ns. MolSA hovered near 750–800 Å^2^, SASA increased from 250–400 Å^2^, and PSA fluctuated between 360–390 Å^2^. A late increase in ligand RMSD was also observed for the reference inhibitor acarbose in the α-glucosidase system, where RMSD rose from about 1.0–2.0 Å to roughly 7.0–8.5 Å towards the end of the trajectory (Appendix A), indicating that both ligands can adopt alternative binding modes under the simulation conditions.

In the peptide 1–α-amylase complex, the backbone RMSD stabilises around 1.4–1.6 Å after initial equilibration (Figure 3B), and most residues show low flexibility (RMSF 0.3–0.9 Å), with a local maximum above 2.7 Å near residues 250–270, indicating a flexible loop or terminal region (Figure 3E). The peptide exhibits higher mobility than the protein, with ligand RMSD fluctuating between 2.5 and 3.5 Å and RMSF values above 2.5 Å at the N- and C-termini. In comparison, the central region remains stable at about 1.0 Å, suggesting an anchored core. The complex’s overall RMSD stabilises around 3 Å, radius of gyration remains between 6.6 and 6.8 Å, and MolSA, SASA, and PSA stay steady, reflecting a compact, solvent-interacting structure (Figure 3H,K). Compared to acarbose bound to α-amylase, with backbone RMSD 1.2–1.3 Å and ligand RMSD 1.0–1.8 Å (Appendix A), peptide 1 shows similar protein stability but higher ligand RMSD (2.5–3.5 Å) and RMSF, indicating a more dynamic binding mode. In contrast, the inhibitor has a more rigid pose.

For the peptide 1–aldose reductase complex, the protein backbone remains stable, with RMSD settling between 1.5 and 1.8 Å after the initial phase (Figure 3C). The ligand RMSD is larger, stabilising around 6.0–6.5 Å with a transient dip below 2.0 Å, indicating pronounced motion within the binding pocket while remaining associated with the protein. Protein RMSF (Figure 3F) confirms a stable core (most residues < 1.0 Å) and higher flexibility around residues ~120–140, ~210–230 and ~290–310 (up to ~3.0 Å), with the most significant fluctuations near the C-terminus (~310). Ligand RMSF (Figure 3I) shows high flexibility of the N-terminal atoms 1–10 (4.0–6.0 Å), lower values of about 1.0–2.5 Å in the middle region (atoms 20–50), consistent with stable contacts in the pocket, and slightly elevated flexibility for atoms 51–57 at the C-terminus. Additional descriptors indicate moderate ligand RMSD variations of 2.0–4.0 Å around the predominant pose, a ligand radius of gyration confined to 6.8–7.5 Å, 1–3 intra-ligand hydrogen bonds, MolSA of about 740–780 Å^2^, SASA between 460 and 580 Å^2^ and PSA in the 320–360 Å^2^ range, all pointing to a compact ligand with limited changes in surface exposure (Figure 3L). In comparison with quercetin bound to aldose reductase, which shows lower ligand RMSD values (up to just above 3.0 Å) and modest atomic RMSF (Appendix A), peptide 1 adopts a more mobile binding pose. However, the protein backbone stability is similar in both systems.

Among the enzyme targets, the peptide 1–α-amylase complex is the most dynamically stable system, displaying low protein RMSD, limited residue fluctuations, and moderate ligand RMSD, which indicates effective peptide anchoring within the catalytic cleft. This stability correlates with α-amylase’s deep, well-defined active site, which facilitates stable peptide binding and limits ligand movement. For α-glucosidase, both peptides exhibit moderate flexibility and ligand mobility; occasional increases in ligand RMSD are followed by re-stabilisation, signalling partial unbinding and rebinding rather than permanent dissociation. This pattern underscores reversible inhibition, which is often only observable in MD simulations rather than in static docking. In aldose reductase complexes, the protein backbone remains stable. Still, the peptides exhibit greater ligand mobility than α-amylase, reflecting the more flexible, solvent-accessible binding site of aldose reductase, which permits ligand movement without compromising overall stability.

For peptide 2, 100 ns molecular dynamics simulations were carried out for its complexes with α-glucosidase, α-amylase, aldose reductase and RAGE. Protein backbone RMSD and RMSF, together with ligand RMSD, RMSF, radius of gyration and surface descriptors (MolSA, SASA, PSA), are summarised in Figure 4 and provide an overview of the stability and flexibility of these complexes in comparison with peptide 1 (Figure 3) and with the reference inhibitors (Appendix A).

In the peptide2–α-glucosidase complex, the protein shows moderate flexibility, with backbone RMSD stabilising between 2.2 and 2.8 Å after the initial phase, without signs of unfolding (Figure 4A). The ligand RMSD rises markedly between 40 and 60 ns to values above 4.5 Å. Then it decreases after 85 ns to about 2.3 Å, indicating transient partial dissociation followed by stabilisation in a new pose. Protein RMSF (Figure 4E) is mostly below 1.0 Å, with more flexible segments above 2.0 Å near residues ~140, ~240, ~410 and at the termini, while regions around ~300–350 and ~450–500 fluctuate only slightly. Ligand atom RMSF (Figure 4I) exceeds 4.0 Å for peripheral atoms, likely solvent-exposed, but remains below 1.5 Å in a central core that appears to maintain stable contacts in the binding site. The complex RMSD stays around 2.5–3.0 Å and the radius of gyration near 6.6 Å (Figure 4M), indicating a compact and globally stable complex despite local protein flexibility and pronounced ligand mobility. Compared with peptide 1 and acarbose bound to α-glucosidase, peptide 2 displays a similarly stable protein backbone but a more clearly reversible ligand rearrangement rather than a predominantly late dissociation event.

In the peptide 2–α-amylase complex, the protein remains highly stable, with backbone RMSD starting at about 1.2 Å and mainly staying between 1.4 and 1.6 Å (Figure 4B). Most residues have RMSF values below 1.0 Å, and only local peaks above 2.7 Å around residues ~110, ~150, ~230, ~300 and ~390 indicate flexible loops or exposed regions (Figure 4F). In contrast, the ligand shows pronounced conformational changes (Figure 4F): its RMSD, initially low, exhibits several fluctuations and a sharp increase above 13 Å near the end of the 100 ns trajectory, pointing to reduced binding stability. Ligand atomic RMSF ranges from about 2.5 to more than 8 Å, with the highest values at the termini and lower values in the central segment, which suggests a more stable core region (Figure 4J). Over the extended simulation window (Figure 4N), the complex RMSD remains between 2.5 and 4.0 Å, the radius of gyration fluctuates around 7.2 Å with a slight decrease, intramolecular hydrogen bonds in the ligand increase over time, and MolSA is essentially constant, while SASA and PSA gradually increase, reflecting enhanced solvent exposure. Overall, the complex retains a stable protein core. Still, peptide 2 is clearly less stably bound to α-amylase than peptide 1 and acarbose, which both show lower ligand RMSD values in their respective complexes (about 2.5–3.5 Å for peptide 1 and 1.0–1.8 Å for acarbose; Appendix A).

In the peptide 2-aldose reductase complex, the protein backbone RMSD increases from 1.5 to 2.0 Å at the beginning and then remains stable (Figure 4C). The ligand RMSD rises rapidly during the first 10 ns. Then it stabilises between 6.0 and 8.0 Å, indicating substantial motion or positional shifts within the binding pocket without an apparent drift towards complete dissociation. Residue-level RMSF (Figure 4G) shows that most residues fluctuate by less than 1.0 Å, while pronounced peaks near residues ~30, ~90, ~130 and 220–230, some above 3.5 Å, mark flexible loop or solvent-exposed regions; α-helices and β-strands fluctuate less. Ligand atom RMSF (Figure 4K) ranges from 1.5 to more than 6.0 Å, with the highest flexibility at the N-terminal region, lower values in the central portion, consistent with more stable contacts, and moderate flexibility at the opposite end. A complementary RMSD analysis indicates moderate complex RMSD fluctuations between 2.5 and 4.5 Å, a protein radius of gyration of 7.0–7.5 Å, fluctuating intramolecular hydrogen bonds, MolSA of 770–800 Å^2^, and an initially increasing, then stable SASA and PSA between 500 and 570 Å^2^ (Figure 4O). These data indicate a structurally stable protein with a mobile peptide 2, which resembles the behaviour of peptide 1 and contrasts with quercetin, whose lower ligand RMSD values (starting at about 0.5 Å and rising only to just above 3.0 Å by the end of the simulation; Appendix A) point to a more tightly anchored binding mode.

In the peptide 2–RAGE complex, the protein backbone RMSD fluctuates between 2.4 and 5.0 Å (Figure 4D), indicating conformational rearrangements while maintaining a folded structure. The ligand RMSD starts below 1.0 Å, then increases after 30 ns, spikes near 65 ns, and exceeds 24 Å by the end, showing extensive displacement and a change in binding mode. RAGE, RMSF (Figure 4H) mostly ranges 1–4 Å, with peaks above 5 Å near residue ~85 and the C-terminus (~210), indicating mobile loops or terminal regions. Segments with RMSF around 1–2 Å are more rigid. Ligand atom RMSF (Figure 4L) ranges from 7 to over 12 Å, higher at terminal atoms and lower in the centre (7–9 Å), suggesting only part of the molecule maintains persistent contacts. The ligand-descriptor plot (Figure 4P) shows internal RMSD around 2.0 Å, radius of gyration about 7.2 Å, and intra-molecular hydrogen bonds between 0 and 6. MolSA remains 740–780 Å2, with SASA and PSA fluctuating slightly around 500–580 Å2. Peptide 2 adopts a compact structure but explores distant positions from RAGE, similar to papaverine, whose ligand RMSD exceeds 50 Å before stabilising at 30–35 Å, and solvent-accessible surface area decreases by roughly 35%, indicating large rearrangements and gradual burial.

Compared to established RAGE inhibitors like Azeliragon (TTP488) and FPS-ZM1, which are low-molecular-weight, drug-like antagonists specifically targeting the extracellular V-domain of RAGE to block ligand binding and signalling, recent studies using crystallography, docking, and molecular dynamics have shown these compounds form stable interactions within shallow, partially enclosed grooves of the V-domain. These interactions are characterised by low RMSD and consistent hydrogen-bond and hydrophobic contacts during simulations [27,30,31]. In contrast, ADVFNPR is a short, food-derived heptapeptide that interacts with RAGE differently: molecular dynamics suggest it maintains a compact structure while showing significant mobility relative to the protein surface, consistent with transient and reversible interactions with the flexible, solvent-exposed V-domain known for recognising structurally diverse ligands rather than a single deep pocket [32,33]. This dynamic behaviour is typical of peptide–RAGE and AGE-derived ligands, contrasting with the more rigid binding pocket of small-molecule antagonists like FPS-ZM1, whose stable binding during MD aligns with their role as systemic RAGE blockers [32]. Therefore, ADVFNPR might be considered a low-affinity, reversible RAGE modulator rather than a direct small-molecule analogue of Azeliragon or FPS-ZM1, with potential relevance for reducing excessive AGE–RAGE signalling during metabolic stress, consistent with current structural and computational insights into RAGE–ligand recognition.

The peptide 2–RAGE complex exhibited the most significant ligand displacement during molecular dynamics simulations, with ligand RMSD values exceeding 24 Å. Such large RMSD values indicate extensive repositioning relative to the initial docking pose and are generally interpreted as weak or transient binding rather than simulation artefacts, particularly when protein secondary and tertiary structures remain stable [34,35]. Importantly, peptide 2 retained a compact internal conformation throughout the trajectory, as evidenced by a stable radius of gyration and persistent intramolecular hydrogen bonding. This observation suggests that the high RMSD arises from the peptide’s translational freedom on the protein surface rather than from peptide unfolding or structural instability.

This behaviour corresponds to current understanding of the RAGE receptor, which binds various ligands through shallow, flexible, and solvent-exposed interfaces, rather than through deep catalytic pockets [33,34]. Consequently, ligand engagement with RAGE often entails dynamic, low-affinity, and multivalent interactions, allowing ligands to probe different surface areas during MD simulations. The observed instability in the peptide 2–RAGE complex likely reflects surface-mediated or atypical interactions rather than stable allosteric inhibition. While RMSD analysis alone cannot conclusively identify allosteric modulation, the absence of a consistent binding configuration suggests that peptide 2 is unlikely to serve as a potent RAGE inhibitor. However, it may transiently interact with the receptor (could be a modulator).

This analysis primarily uses structural stability metrics, such as RMSD and RMSF, to understand peptide–RAGE interactions. While methods such as Molecular Mechanics-Generalised Born Surface Area (MM/GBSA) or Molecular Mechanics- Poisson Boltzmann Surface Area (MM/PBSA) could offer additional thermodynamic insights, they were outside the current study’s scope. They may be explored in future work to better quantify peptide–RAGE binding energetics [36]. Additionally, although MD simulations indicate that peptide 2 maintains a compact structure, experimental validation by circular dichroism (CD) or nuclear magnetic resonance (NMR) spectroscopy will be necessary to confirm these conformational preferences under solution conditions.

Analysis across all targets indicates that peptide 1 generally demonstrates greater dynamic stability than peptide 2. In enzyme complexes, peptide 1’s ligand RMSD usually remains between 2–7 Å, whereas peptide 2 shows larger fluctuations, exceeding 13 Å in the α-amylase system and over 24 Å in the RAGE simulation. During MD simulations, a steady, moderate ligand RMSD typically reflects stable binding with minor conformational shifts. Conversely, sudden RMSD spikes imply loss of the original binding pose or the presence of transient interactions. The consistently lower mobility of peptide 1 suggests it binds more strongly and maintains interactions more effectively than peptide 2.

### 2.5. Peptide Structure Prediction

Before structural prediction, the two lead peptides identified by nano-ESI-Orbitrap-LC-MS/MS were synthesised and analytically confirmed. Reverse-phase HPLC showed a single sharp peak for each peptide, with retention times of 10.018 and 11.702 min for peptide 1 (MQIFVK) and peptide 2 (ADVFNPR), respectively, consistent with purity above 95% (Appendix A). Electrospray mass spectra confirmed the expected molecular masses, with peptide 1 exhibiting a single charge state at *m*/*z* 765.5 [M + H]^+^ and peptide 2 exhibiting charge states at *m*/*z* 818.5 [M + H]^+^ and 409.9 [M + 2H]^2+^ (Appendix A).

To provide atomistic insight into their conformational preferences and binding-competent geometries, we next applied de novo structure prediction using the PEP-FOLD 4.0 approach. The final output includes the five best-ranked models, based on energy and clustering metrics, along with an archive of all generated models. The probability plots for both peptides are shown in Figure 5. The top 5 models of both peptides are depicted in Figure 6.

Figure 5 shows PEP-FOLD probability-based Ramachandran-like density plots that summarise the conformational preferences of the two peptides. These visualisations display the conformational tendencies of residues in each peptide, with colour coding typically indicating the likelihood of secondary structure: red indicates an α-helix, blue a β-sheet, and green a coil or random structure. For peptide 1, residues M, Q, and I display significant red areas (below 0 0.2 on the y-axis), especially near the bottom, indicating a strong α-helical tendency at the N-terminal region. A shift to green and light green in the middle to the upper areas (above 0.3, 3) suggests coil or less-defined structures. The minimal blue regions imply that β-sheet formation is not the main structure in this peptide. Peptide 1 shows partial α-helical character in its N-terminal residues due to amino acids M-Q-I, which transition into more flexible coil regions further along the sequence. This indicates that the peptide may start folding by forming a helix, but does not adopt a stable secondary structure. The ability to form helices can enhance receptor binding or membrane penetration, particularly when involved in bioactive processes [37]. For peptide 2, the plot is predominantly blue, especially across the D, V, and F residues, indicating a strong tendency to form β-sheets. Notably, residue D shows a distinct red segment at the lower end (around 0.2, 2), suggesting some propensity toward alpha-helix formation. Peptide 2 shows a strong tendency to form β-sheets, particularly among its hydrophobic and aromatic residues, suggesting a more compact and stable structure. The presence of Arg (R) and Asp (D) enables electrostatic and hydrogen-bonding interactions that further stabilise its conformation. The scattered red and green areas indicate minor conformational flexibility, but do not suggest substantial helical content. Its overall β-rich structure may provide resistance to aggregation, enhance receptor specificity, or increase bioactivity, aligning with its high sOPEP scores.

The five lowest-energy structures predicted by PEP-FOLD for each peptide (see Figure 6) represent the most probable solution-state conformations before binding to the target. These structures are key to understanding their biological function. For peptide 1 (MQIFVK), the five models exhibit notable diversity, with differences in backbone shape and hydrophobic side-chain orientations, suggesting a flexible, broad conformational range. This flexibility supports effective enzyme inhibition, especially for enzymes with defined catalytic pockets, where induced-fit mechanisms are typical. This aligns with peptide 1’s activity against α-glucosidase, α-amylase, and aldose reductase. However, this heterogeneity also entails a higher entropic cost during binding and less consistency across different targets, potentially limiting its wider effectiveness. In contrast, peptide 2 (ADVFNPR) exhibits high similarity across its five conformations, all adopting compact structures with a conserved backbone, suggesting a low-energy region. This stability results from interactions among charged residues (Asp, Arg), polar residues (Asn), conformationally restricted residues (Pro), and aromatic residues (Phe), which promote intramolecular electrostatic interactions and backbone rigidity. Such a pre-organised yet adaptable ensemble can facilitate multi-target binding by lowering the entropic barrier while maintaining sufficient flexibility for protein adaptation consistent with conformational ensemble and selection models [38,39]. This structural characteristic explains why peptide 2 could exhibit higher biological activity, including lower IC_50_ values, competitive inhibition, and a broader target range, as observed experimentally. Molecular dynamics simulations support this, showing peptide 2 remains structurally compact internally while adjusting its position on protein surfaces, especially with RAGE. This behaviour aligns with models of dynamic molecular recognition, in which ligands transiently interact with flexible, solvent-exposed surfaces rather than rigid pockets [33,36]. Notably, the large RMSD values for peptide 2 in the RAGE complex reflect surface exploration rather than unfolding, a common observation in MD studies of peptides with moderate or weak affinity [35]. Overall, the five conformations suggest that peptide 2 might be a stable, pre-organised structure capable of supporting reversible, adaptable interactions, which might account for its multifunctional bioactivity. In contrast, peptide 1’s greater flexibility might correlate with effective enzyme inhibition but a more limited interaction profile.

### 2.6. Safety Assessment of Coconut Milk Peptides by Hemolytic Assay

The hemolytic activity of the peptides was assessed using their HC_50_ values (Table 4). Erythrocyte-based hemolysis assays are widely used as an early, quantitative screen for peptide-associated hemotoxicity, with higher HC_50_ values indicating lower hemolytic potential [40]. Peptide 2 exhibited a lower hemolytic potential than peptide 1, with HC_50_ values of 231.53 ± 0.14 µg/mL and 220.85 ± 0.20 µg/mL, respectively. The positive control, Triton X-100, showed a markedly lower HC_50_ (105.25 µg/mL), consistent with strong membrane-disrupting activity. Overall, the higher HC_50_ values for both peptides indicate substantially weaker disruption of erythrocyte membranes than Triton X-100. Importantly, hemolytic activity is a key safety liability that has been linked to systemic in vivo toxicity of membrane-active peptides [41]. Therefore, the lower hemolytic activity of peptide 2 supports its prioritisation for further pharmacological evaluation. In summary, peptide 2 appears to be the safer choice for erythrocyte compatibility.

**Table 3 ijms-27-00360-t003:** Enzyme kinetics of α-glucosidase, α-amylase, and aldose reductase enzymes by Peptide 2.

Enzymes	Treatment	Mode of inhibition ^x^	K_m_ (mM)	V_max_ 10^3^(µM/min)^−1^	K_i_(µg) ^y,z^
α—Glucosidase	Control	Competitive	2.26	15.15	0.95 ± 0.09
	IC_20_—13.13 µg		1.78	14.87	
	IC_40_—26.26 µg		0.60	14.68	
	IC_60_—39.39 µg		0.18	14.50	
α—Amylase	Control	Competitive	3.15	28.28	1.34 ± 0.12
	IC_20_—33.55 µg		2.00	27.75	
	IC_40_—67.10 µg		0.79	26.26	
	IC_60_—100.60 µg		0.22	27.50	
Aldose reductase	Control	Competitive	6.30	47.35	0.78 ± 0.05
	IC_20_—8.12 µg		4.86	46.64	
	IC_40_—16.23 µg		2.02	46.02	
	IC_60_—24.35 µg		0.66	45.55	

^x^ inhibition mode was determined from the Lineweaver–Burk plot. ^y^ K_i_ = apparent inhibition constant. ^z^ Values are expressed as mean ± SE.

Although data from in vitro and in silico studies show that these peptides derived from coconut milk have strong multi-target effects, the issue of oral bioavailability for food-derived peptides remains essential. When taken orally, peptides may be partially broken down or completely broken down into amino acids by gastrointestinal enzymes and often have limited ability to cross the intestinal lining due to their polarity and susceptibility to enzymatic degradation. However, the peptides identified here are short, comprising 6 to 7 amino acids, and were produced using a simulated gastrointestinal digestion method. This suggests that peptides might survive in the intestinal lumen after oral consumption. However, further studies are needed to prove its oral bioavailability and permeability.

### 2.7. Evaluation of In Vitro Antihyperglycemic, Anti-Glycation Activities of Peptides and Enzyme Inhibition Kinetics of the Lead Peptide

In vitro assays were conducted to evaluate whether the two coconut milk peptides can inhibit key hyperglycemia-related enzymes and scavenge the reactive dicarbonyl methylglyoxal. The results were compared with standard drugs (acarbose, quercetin, and creatine) and with previously reported food-derived peptides to assess the relative potency of the coconut peptides and their potential as multi-target anti-hyperglycemic and anti-glycation agents. The results of the α-glucosidase, α-amylase, and aldose reductase inhibition assays (IC_50_ values) are tabulated in Table 4.

Among the test compounds, peptide 2 demonstrated the most potent inhibition against yeast α-glucosidase. The IC_50_ values were 38.98 ± 0.23 µg/mL (51.02 µM) for peptide 1 and 32.82 ± 0.17 µg/mL (40.12 µM) for peptide 2, both lower than that of the standard drug acarbose (46.16 ± 0.23 µg/mL; 71.49 µM). These values are markedly more potent than those reported by Wang et al. [42], where soy protein-derived peptides LPLPVLK, SWLRL and WLRL inhibited α-glucosidase with IC_50_ values of 237.43 ± 0.52, 182.05 ± 0.74 and 162.29 ± 0.74 µmol/L, respectively. Compared with other plant-based milk peptides, coconut milk peptides also showed superior inhibitory activity: oat-derived peptides reported by Darewicz et al. [23] and Rafique et al. [43] inhibited α-glucosidase with IC_50_ values typically between 110 and 185 µM, whereas peptide 1 and peptide 2 from coconut milk exhibited IC_50_ values of 40.12 and 51.02 µM, respectively. Concerning the α-amylase inhibition assay, peptide 2 displayed the highest inhibition against α-amylase. The IC_50_ values for peptides 1 and 2 were 88.25 ± 0.34 μg/mL (115.36 μM) and 83.88 ± 0.22 μg/mL (102.55 μM), respectively (Table 4). However, the standard drug, acarbose, exhibited an IC_50_ of 89.62 ± 0.46 μg/mL (138.81 μM). The findings presented in this study are consistent with the study by Patil et al. [44], who reported that a whey-protein hydrolysate from bovine colostrum inhibited α-amylase with an IC_50_ of 82.43 ± 0.09 µg/mL, which was slightly lower than the value reported for acarbose (91.37 ± 0.22 µg/mL). The present results, therefore, suggest that coconut milk peptides may display inhibitory potency comparable to bioactive peptides derived from animal milk. In contrast, González-Montoya et al. [45] reported α-amylase inhibition by peptides from germinated soybeans with an IC_50_ of 1.70 mg/mL (1700 µg/mL). Similarly, Admassu et al. [46] described α-amylase inhibition by red seaweed (*Porphyra* spp.) peptides, Gly-Gly-Ser-Lys (GGLK) and Glu-Leu-Ser (GLS), with IC_50_ values of 2.58 ± 0.08 mM and 2.62 ± 0.05 mM, respectively. Although direct quantitative comparisons are limited by differences in assay conditions and concentration units, these literature values indicate weaker inhibition than that observed for the coconut milk peptides in the present study. Collectively, the results support the use of coconut milk peptides as promising α-amylase inhibitors relevant to postprandial glycaemic control. Peptide 2 effectively inhibited aldose reductase. The IC_50_ value of peptide 1 and peptide 2 was found to be 22.51 ± 0.04 μg/mL and 20.29 ± 0.02 μg/mL, respectively. However, the standard drug, quercetin, exhibited an IC_50_ of 25.22 μg/mL. The results of the current study are comparable to those of previous research by Maradesha et al. [47], where the polyphenol rutin from the whole jackfruit flour inhibited the aldose reductase with the IC_50_ values of 7.86 ± 0.33 μg/mL, and the positive control acarbose exhibited the IC_50_ value of 11.05 ± 0.23 μg/mL. Although the polyphenol exhibited better activity compared to peptides, the coconut milk-derived peptides also exhibited better activity, suggesting that coconut milk bioactive peptides could be an anti-hyperglycemic agent, specifically an aldose reductase inhibitor. Another study by Patil et al. [44] demonstrates that the whey protein hydrolysate fraction from bovine colostrum inhibits human aldose reductase with an IC_50_ of 51.296 ± 0.26 µg/mL. In contrast, the control drug, zenarestat, has an IC_50_ of 59.68 ± 0.47 µg/mL, which is lower than that of coconut milk peptides. Although there is limited literature on peptide-based inhibition of aldose reductase, some plant-derived extracts and polyphenols have been characterised. For instance, dietary extracts from vegetables and spices demonstrated AR-inhibition with IC_50_ values of around 200 µg/mL [47,48], whereas the present peptides (20 µg/mL) were roughly ten times more potent. Recent articles highlight growing interest in natural AR inhibitors, such as synthetic analogues, polyphenols, and functional food peptides, as potential new approaches for managing diabetic complications. Among these, coconut-milk peptides could be an effective food-derived AR inhibitor.

Based on its consistently low IC_50_ values against α-glucosidase, α-amylase and aldose reductase (Table 3), together with a favourable docking profile and good haemocompatibility, peptide 2 (ADVFNPR) was selected as the lead candidate for detailed kinetic analysis. Lineweaver–Burk plots (Appendix A) for all three enzymes showed families of straight lines with similar Y-intercepts and progressively shifted X-intercepts as peptide 2 concentration increased. Consistent with this behaviour, increasing peptide concentration decreased the apparent K_m_ values for α-glucosidase from 2.26 to 0.18 mM, for α-amylase from 3.15 to 0.22 mM, and for aldose reductase from 6.30 to 0.66 mM. At the same time, V_max_ remained essentially constant at 15, 27 and 46 µM min^−1^, respectively (Table 3). This pattern indicates a reversible inhibition that primarily affects apparent substrate affinity rather than catalytic turnover. The apparent K_i_ values (0.95 ± 0.09, 1.34 ± 0.12 and 0.78 ± 0.05 µg mL^−1^) confirm a high affinity of peptide 2 for the enzyme binding sites and agree with the docking-based binding modes. These results are consistent with previous reports on food protein hydrolysates that reversibly inhibit α-glucosidase, α-amylase, and aldose reductase, and mimic the structural features of their natural substrates.

Concerning the LB plot of α-glucosidase (Appendix A), it demonstrates a series of straight lines with similar Y-intercepts and progressively shifted X-intercepts as peptide 2 concentration increases. As inhibitor concentration increases (IC_20_–IC_60_ µg), the apparent K_m_ decreases markedly (from 2.26 to 0.18 mM), whereas V_max_ remains nearly constant (≈15 µM min^−1^). This pattern indicates a reversible inhibition that mainly affects apparent substrate affinity rather than catalytic turnover. The calculated apparent K_i_ (0.95 ± 0.09 µg mL^−1^) indicates a strong affinity between peptide 2 and the binding region of α-glucosidase, possibly mimicking glucose or its analogues.

For α-amylase, the LB plot also shows families of straight lines with closely grouped Y-intercepts. With increasing inhibitor concentration, apparent K_m_ decreases from 3.15 to 0.22 mM, while V_max_ remains relatively constant (≈27 µM min^−1^). The evident K_i_ (1.34 ± 0.12 µg mL^−1^) further supports potent reversible inhibition. Peptide 2 is likely to interact with starch-derived substrates at the catalytic region, for example, via hydrogen bonding with residues such as Asp197, Glu233 and Asp300 [49]. The LB plot of aldose reductase inhibition (Appendix A) similarly reveals families of lines with comparable Y-intercepts and shifting X-intercepts as inhibitor levels increase. Apparent K_m_ decreases significantly (from 6.30 to 0.66 mM) while V_max_ remains steady (≈46 µM min^−1^), and the evident K_i_ value (0.78 ± 0.05 µg mL^−1^) denotes high inhibitory potency. These features are compatible with a reversible inhibition mechanism that modulates substrate binding at or near the NADPH-dependent catalytic site without substantially affecting catalytic turnover, in agreement with the behaviour reported for several flavonoid- and peptide-type inhibitors occupying the anionic binding pocket near Tyr48, His110 and Trp111 [48]. Whey protein hydrolysate (WP) from bovine colostrum has likewise been reported to inhibit all three diabetic targets, exhibiting an inhibition pattern comparable to that observed in this study. The inhibition of these targets by peptides or hydrolysates is attributed to their structural similarity to the original substrates [44].

### 2.8. Evaluation of Methyl Glyoxal Scavenging Activity of Peptides

The methylglyoxal scavenging results (EC_50_) are summarised in Table 4, alongside the results of the remaining bioactivity and safety assays. Peptide 2 showed the lowest EC50 at 37 °C with 2 h of incubation (182.17 ± 0.23 µg/mL), followed by peptide 1 (190.13 ± 0.50 µg/mL). In contrast, the standard drug creatine reached 230.05 ± 0.34 µg/mL. Risum et al. [50] and Zhu et al. [51] incubated methylglyoxal with BSA or a scavenger peptide for 2 h at 37 °C, and these studies formed the basis for the modified preliminary Maillard reaction model used here. The present EC_50_ values are in line with previous reports on methylglyoxal-scavenging peptides: Deng et al. [52] described a *Ginkgo biloba* peptide (VVFPGCPE) that achieved up to 66.50% scavenging within 24 h at 5 mg/mL, whereas the coconut milk peptides reached up to 50% scavenging after 2 h at 37 °C with an EC50 of 182.17 ± 0.23 µg/mL. This activity is consistent with the presence of an arginine residue, which is known to react with dicarbonyls such as methylglyoxal [53]. Another study by Brings et al. [54] reported that an arginine-rich, fatty acid-coupled cyclic peptide, CycK(Myr)R4E, scavenged methylglyoxal with EC50 values of 110.95 µg/mL (100 µM) and 443.95 µg/mL (400 µM). Although the lower concentration of CycK(Myr)R4E (110.95 µg/mL) is more potent than the coconut peptide (182.17 µg/mL), both EC_50_ values remain below 200 µg/mL and are therefore comparable. In addition, both studies used creatine as a positive control. Collectively, these data indicate that coconut milk peptides possess significant methylglyoxal-scavenging capacity and may help reduce dicarbonyl stress, thereby limiting non-enzymatic glycation under diabetic conditions.

**Table 4 ijms-27-00360-t004:** Outcomes of anti-hyperglycemic, anti-glycation, safety assessment and anti-oxidant assays.

Samples	IC_50_ (µg/mL/µM) / EC_50_ (µg/mL/µM)								
	Anti-Hyperglycemic Assays			Anti-Glycation Assay (Methylglyoxal Scavenging)		Safety Assessment of Peptides: Hemolytic Assay (HC_50_) (µg/mL/µM)	Anti-Oxidant Assays		
	α-Glucosidase Inhibition	α-Amylase Inhibition	Aldose Reductase Inhibition	EC_50_ (µg/mL/µM) at RT	EC_50_ (µg/mL/µM) at 37 °C, 2 h		DPPH Radical Scavenging Assay	ABTS Radical Scavenging Assay	Superoxide Radical Scavenging Assay
Peptide 1	38.98 ± 0.23 ^b^ (50.96)	88.25 ± 0.34 ^b^ (115.36)	22.51 ± 0.04 ^b^ (29.43)	215.17 ± 0.44 ^b^ (281.28)	190.13 ± 0.50 ^b^ (248.54)	220.85 ± 0.20 ^b^ (288.70)	26.14 ± 0.12 ^c^ (34.17)	27.89 ± 0.17 ^c^ (36.46)	52.32 ± 0.30 ^b^ (68.39)
Peptide 2	32.82 ± 0.17 ^a^ (40.13)	83.88 ± 0.22 ^a^ (102.56)	20.29 ± 0.02 ^a^ (24.81)	208.10 ± 0.65 ^a^ (254.44)	182.17 ± 0.23 ^a^ (222.73)	231.53 ± 0.14 ^c^ (283.08)	22.14 ± 0.12 ^a^ (27.07)	17.89 ± 0.20 ^a^ (21.87)	42.45 ± 0.30 ^a^ (51.90)
* Acarbose	46.16 ± 0.23 ^c^ (71.50)	89.62 ± 0.46 ^c^ (138.82)	-	-	-	-	-	-	-
** Quercetin	-	-	25.22 ± 0.07 ^c^ (83.44)	-	-	-	-	-	-
*** Creatine	-	-	-	230.05 ± 0.34 ^c^ (1754.37)		-	-	-	-
**** Triton-X-100	-	-	-	-	-	105.25 ± 0.17 ^a^ (162.67)	-	-	-
***** Ascorbic acid	-	-	-	-	-	-	28.65 ± 0.17 ^b^ (162.67)	22.74 ± 0.27 ^b^ (129.12)	57.71 ± 0.16 ^c^ (327.67)

Values are expressed as mean ± SD. Means in the same column with distinct superscripts (a–c) are significantly different (*p* ≤ 0.05) as separated by the Tukey test. Peptide 1—Coconut milk peptide 1; peptide 2—Coconut milk peptide 2. * Acarbose was used as a positive control for α-glucosidase inhibition and α-amylase inhibition assay, ** Quercetin was used as a positive control for aldose reductase inhibition assay, *** Creatine was used as a positive control for the methylglyoxal scavenging assay, **** Triton-X-100 was used as a positive control for the hemolytic assay and ***** Ascorbic acid was used as a positive control for the anti-oxidant assays. Note—The values in the bracket are in micromolar (µM).

### 2.9. Evaluation of the Antioxidant Potential of Coconut Milk Peptides

The results of the antioxidant assay are represented as EC_50_ values tabulated in Table 4 along with other outcomes of bioassays. Peptide 2 demonstrated the highest antioxidant activity across all evaluation methods. Concerning the DPPH and ABTS radical scavenging, the results of the previous study by Wang et al. [55] are not in par with the current research. In a previous study, peptides obtained from the *Gossypium hirsutum* by-product exhibited DPPH scavenging activity with EC_50_ values of 2.05 ± 0.02 mg/mL. The same peptides also scavenged the ABTS cation radical, with EC_50_ values of 0.49 ± 0.02 mg/mL. The outcomes suggest that coconut milk peptides exhibit greater antioxidant activity than previously reported.Regarding superoxide radical scavenging using the NBT method, the results do not align with those of the previous study by Chen et al. [56] wherein the two peptides LLLRW and GDMNP from *Oryza sativa* scavenged the superoxide anion radical with the EC_50_ values of 0.430 ± 0.012 mg/mL and 0.400 ± 0.008 mg/mL, respectively. Suetsuna et al. [57] reported a casein-derived peptide YFYPEL with superoxide scavenging activity of 65.8 µg/mL (79.2 µM), which is slightly less potent compared to the coconut milk peptides. The outcomes of the present study are consistent with those of the previous study by Patil et al. [44] in which the whey protein hydrolysate (WP) from bovine colostrum scavenged the DPPH, ABTS and superoxide radicals with the EC_50_ of 38.241 ± 0.23 μg/mL, 42.819 ± 0.33 μg/mL, and 69.897 ± 0.10 μg/mL, respectively. The outcomes were comparable and slightly higher (lower potency, as indicated by the EC_50_ values). The WP hydrolysate fraction from previous studies exhibited higher EC_50_ values than those in the current study, suggesting that coconut milk-derived peptides may be more effective antioxidants, thereby decreasing oxidative stress generated through T2DM-linked AGE-RAGE signalling.

## 3. Materials and Methods

### 3.1. Materials

Matured coconuts, centrifuge tubes, microcentrifuge tubes, spatula, analytical weighing balance, plastic container, 3 kDa ultrafiltration filters (0.5 mL), micropipettes, microtips, ice flakes, mini-cooler, and ice tray.

Sodium hydroxide [≥98%, Sisco Research Laboratories (SRL), Mumbai, Maharashtra, India], pepsin (from porcine gastric mucosa, lyophilised powder, ≥2500 units/mg protein, SRL), pancreatin (from porcine pancreas, 8x USP specification, SRL), sodium azide (NaN_3_, ≥99.5%, SRL), sodium phosphate dibasic anhydrous (≥99.5%, Merck, Darmstadt, Germany), sodium phosphate monobasic (≥99.5%, Sigma-Aldrich, Darmastadt, Germany), boric acid (SRL), potassium chloride (KCl, 99.5%, SRL), acetic acid (Thermo Fisher, Waltham, MA, USA), urea (≥99.5%, Sigma-Aldrich), nicotinamide adenine dinucleotide phosphate reduced (NADPH, extrapure 98%, SRL), ammonium sulphate (≥99%, SRL), methylglyoxal (MGO, 40% *w*/*v*, Sigma-Aldrich), 2,4-dinitrophenyl hydrazine (DNPH, 97%, Sigma-Aldrich), BCA Kit (Sigma-Aldrich), sodium sulfite (SRL), Bradford reagent (HiMedia, Mumbai, Maharashtra, India), acrylamide-bisacrylamide (30%, BioRad, Hercules, CA, USA), Tris buffer (pH 6.8, 0.5 M and pH 8.8, 1.5 M, SRL), ammonium persulfate (Sigma-Aldrich), sodium dodecyl sulphate (SDS, ≥99.5%, Sigma-Aldrich), tetramethyl ethylenediamine (TEMED, Sigma-Aldrich), Triton-X-100 (SRL), Coomassie Brilliant Blue (CBB) G250 (SRL), C18 discs (Empore, Sigma-Aldrich), trypsin (3x crystal, extracted from bovine pancreas, SRL), β-NADPH (SRL), 2,2-diphenyl-1-picrylhydrazyl (DPPH, Sigma-Aldrich), 2,2′-Azino-bis(3-ethylbenzothiazoline-6-sulfonic acid (ABTS) (Sigma-Aldrich), Nitroblue tetrazolium (NBT, Sigma-Aldrich), phenazine methosulphate (PM, Sigma-Aldrich), aminoguanidine (Sigma-Aldrich), porcine pancreatic α-amylase (EC 3.2.1.1, SRL), α-glucosidase (EC 3.2.1.20, SRL) from *Saccharomyces cerevisiae*, p-nitrophenyl-α-d-glucopyranoside (pNPG, SRL), and nicotinamide adenine dinucleotide-reduced (NADH, SRL). Ultrapure water, obtained from a Milli-Q purification unit (Millipore, Bedford, MA, USA), was used throughout the process.

### 3.2. Extraction, Estimation, and Profiling of Coconut Milk Proteins

Mature coconuts were purchased from the local farm and thoroughly washed. Coconut milk was extracted using the traditional method. Initially, the mature coconuts were broken and subsequently grated using a coconut grater (Wise Innovations 220 V, Daman, India). Further, the coconut gratings were blended with a small volume of water (around 50–100 mL). Subsequently, the blended coconut gratings were passed through clean cheesecloth and manually squeezed, yielding a milky-white liquid, i.e., coconut milk [58].

The extracted coconut milk was centrifuged at 8000 rpm for 40 min, at 4 °C (REMI refrigerated centrifuge RM 12C, Mumbai, India). Three separate fractions were obtained: coconut cream (top fraction), skimmed coconut milk (middle aqueous fraction), and an insoluble pellet (bottom fraction). Furthermore, all three fractions were stored separately at −20 °C until further analysis. Skimmed coconut milk was subjected to dialysis using a three-kDa dialysis membrane, and the retentate obtained after dialysis was collected in a separate centrifuge tube, which served as the protein source [59].

2 g of coconut cream was accurately weighed, resuspended in phosphate-buffered saline (PBS, 0.01 M, pH 7.2), and centrifuged twice at 5000 rpm for 15 min at 4 °C, followed by a wash with water under the same conditions. Washed fat globules of both samples were suspended in distilled water (1:1 *v*/*v*) and allowed to crystallise for 24 h at 4 °C. Following homogenisation, the samples were vortexed (Vortex, Vortex Genie 2 mixer—Sigma-Aldrich, Darmstadt, Germany) to further mix their contents. The obtained fat globule membrane proteins (FGMPs) suspension was centrifuged at 5000 rpm for 15 min at 5 °C to separate the fat and fat globule fractions. The obtained fat was heated to 45 °C for 30 min, then mixed with distilled water. This suspension was centrifuged at 5000 rpm for 15 min at 4 °C to remove traces of fat. The remaining fat globule suspension was combined with the previous one, which served as the protein source for further analysis [60].

1 g of insoluble pellet was weighed, reconstituted with 10 mL of 10 mM PBS, and centrifuged at 5000 rpm for 15 min at 4 °C, and the resulting supernatant was used as the protein source.

The protein content in the coconut milk samples was analysed using the BCA method [61]. In brief, 10 µL of the samples (diluted 1:3) were mixed with 200 µL of BCA reagent and incubated at 37 °C for 30 min. After incubation, the absorbance of the purple complex was recorded at 562 nm, using a blank sample lacking the BCA reagent as the reference. The protein concentration in the samples was determined using a BSA calibration curve (0–2500 µg/mL).

SDS-PAGE was carried out to characterise the protein profile of coconut milk following the method of Laemmli [62] as adapted by Senphan and Benjakul [14]. Protein separation was performed by SDS–PAGE on polyacrylamide gels composed of a 4% stacking layer overlaying a 12% resolving layer. Crude coconut milk protein extracts were mixed with the sample buffer and heated at 90 °C for 10 min before loading. An amount of 50 µg of protein was applied to each lane, and electrophoresis was performed in a vertical gel electrophoresis unit (Mini-Protein II; Bio-Rad Laboratories, Richmond, CA, USA) at a constant voltage of 200 V. After electrophoresis, gels were stained with Coomassie Brilliant Blue R-250 (25% methanol, 10% acetic acid, *v*/*v*). Destaining was performed in 40% methanol and 10% acetic acid (*v*/*v*) until the background was transparent and protein bands were well resolved.

### 3.3. In Vitro Gastro-Intestinal Protein Digestion

The in vitro digestion of coconut milk proteins was performed as described by Patil et al. [60] to simulate gastrointestinal protein digestion under physiologically relevant conditions. In brief, 3.5 g of protein samples were dissolved in 100 mL of KCl-HCl buffer. 96 mL of protein samples were measured, to which 4 mL of 4% pepsin was added.

This protein-protease mixture was thoroughly mixed and incubated at room temperature for 4 h. Following incubation, the reaction was stopped by placing the samples in a boiling water bath for 10 min. Subsequently, the samples were neutralised using 2 N NaOH. Further, the samples were centrifuged at 5000 rpm for 45 min. The supernatants obtained were further hydrolysed with 4% pancreatin and 4% trypsin, as described above.

### 3.4. Identification of Peptides by Nano-LC-MS/MS Orbitrap Analysis

The protein hydrolysates were first desalted with C18 discs and then concentrated by centrifugation at 6000 rpm for 20 min at 4 °C, using three kDa filters (0.5 mL). Subsequently, the desalted and concentrated hydrolysates were lyophilised and sent to SAIF-IIT Bombay for peptide identification.

The liquid chromatography analysis utilised a Thermo EASY-nLC system (ThermoFisher, Waltham, MA, USA). Initially, an eight μL sample was collected at a rate of 5 μL/min. During loading, 12 μL of the sample was used, with a maximum pressure of 750 Bar. The gradient started at 0 min with a flow of 300 nL/min and 2% B, lasting for 5 min. Further, it was increased to 15% B at 80 min, and to 45% B at 100 min. At 105 and 110 min, the mixture B was increased to 95%, then reduced back to 2% at 115 and 120 min, maintaining a flow rate of 300 nL/min throughout. Before analysis, the pre-column was equilibrated with eight μL under a maximum pressure of 600 Bar, while the analytical column was equilibrated similarly at 700 Bar. The autosampler was washed with a 0.1 mL flush to ensure proper cleaning. The mass spectrometry analysis was performed using a Q Exactive Plus Orbitrap MS, Waltham, MA, USA. The method was configured for a 120-minute run, with the chromatographic peak width set to 15 s. Full MS and data-dependent MS/MS (dd-MS^2^) experiments were conducted in positive ion mode over a 0 to 120 min runtime. Full-scan spectra were acquired with one microscan at a resolving power of 70,000 (at *m*/*z* 200), an AGC target of 1 × 10^6^ ions and a maximum injection time of 60 ms, over an *m*/*z* range of 350–2000 in profile mode. For data-dependent MS/MS acquisition, a resolution of 17,500 was used with an AGC target of 2 × 10^5^ ions and a maximum injection time of 120 ms; the 15 most intense precursor ions in each cycle were isolated and fragmented with a normalised collision energy of 27. To minimise repeated selection of identical precursors, dynamic exclusion was applied for 35 s. The mass spectrometry was tuned using the “Nitrosamine_12062023.mstune” file to ensure optimal performance. The pump was configured at a flow rate of 3 μL/min and a volume of 250 μL. Lock masses and inclusion/exclusion lists were not utilised in this method [60,63].

### 3.5. Physico-Chemical Properties of Peptides

After obtaining peptide sequences using the nano-LC-MS/MS Orbitrap technique, duplicate peptides were manually removed. Further, they were analysed for their physicochemical properties, which include molecular weight, bioactivity prediction using peptide ranker, number of amino acids (peptide length: 6–15 amino acids), allergenicity, toxicity, molecular formula, molecular weight (<2.5 kDa), net charge, pI, the total number of positive and negatively charged residues, stability index (<40), extinction coefficient, hydrophobicity and solubility using ProtParam (Expasy, Switzerland) [64], PepCalc (Innovagen AB, Sweden) [65], AlgPred (Indian Institute of Information Technology, Delhi) [66] and ToxinPred (IIIT, Delhi) [67] webtools.

### 3.6. Molecular Docking

The screened peptides were evaluated for their potential antihyperglycemic and antiglycation activities using in silico molecular docking. Docking simulations were performed in Schrödinger’s Biologics Suite against α-glucosidase, α-amylase, aldose reductase, and the receptor for advanced glycation end products (RAGE). The X-ray crystal structures of α-glucosidase (PDB ID: 3A4A; [68]), α-amylase (PDB ID: 1DHK; [69]), aldose reductase (PDB ID: 1IEI; [70]), and RAGE (PDB ID: 3CJJ; [71]) were retrieved from the RCSB Protein Data Bank (PDB, https://www.rcsb.org/, accessed on 2 May 2025). Protein structures were processed in Maestro (v14.3; Schrödinger Release 2024-4) with the Protein Preparation Wizard workflow. In the first step, crystallographic water molecules were removed, and each protein was then subjected to restrained energy minimisation with the OPLS force field, converging heavy-atom positions to an RMSD of 0.30 Å before docking. Receptor grids were generated by defining the ligand-binding sites either around the co-crystallised ligands (for all targets except α-amylase, which lacks a bound ligand) or based on key active-site residues in α-amylase. Peptide ligands were built in Maestro’s 2D Sketcher and subsequently energy-minimised with the OPLS force field. The prepared peptides were then docked into the corresponding receptor grids using the peptide docking protocol implemented in the Biologics module of Maestro 14.3 [72], and docking poses were analysed and visualised in Maestro.

### 3.7. Molecular Dynamics Simulations

Molecular dynamics (MD) simulations were performed using the Desmond simulation package (Schrödinger LLC, New York, NY, USA) under an NPT ensemble at 310 K and 1 bar. For each system, an initial relaxation protocol with a 1 ps relaxation time was applied, followed by 100 ns production runs. All simulations were conducted using the OPLS_2005 force field settings [73]. Long-range electrostatic interactions were calculated with the particle mesh Ewald method, using a real-space Coulomb cutoff radius of 9.0 Å [74]. A simplified point-charge water model, as described by Pathirannahalage et al. [75], was used to represent the solvent. The r-RESPA multiple-time-step scheme was used for non-bonded terms, with both the short- and long-range components refreshed every three integration steps; coordinates were saved at 4.8 ps intervals. Ligand–protein interactions were analysed using the Simulation Interaction Diagram tools implemented in the Desmond MD package, and the stability of the simulations was assessed by monitoring the root mean square deviation (RMSD) of ligand and protein atom positions throughout the 100 ns trajectories.

### 3.8. Peptide Structure Prediction

The best 3D structures of the peptides were predicted using PepFold 4 (https://bioserv.rpbs.univ-paris-diderot.fr/services/PEP-FOLD4/, accessed on 2 May 2025). Initially, peptide sequences were provided in FASTA format. Structure prediction involved three steps: first, the amino acid sequence was subjected to structural alphabet (SA) prediction using an SVM trained on evolutionary profiles from Psi-BLAST.(NCBI, Bethesda, MD, USA) Next, coarse-grained modelling was employed to generate initial structures, guided by the sOPEP energy function, with a Debye–Hückel-based term incorporated to account for electrostatic interactions. These models were then refined to full atomic detail using rotamer libraries for side-chain placement and optimisation. The models were clustered based on structural similarity, and a sorting algorithm ranked them to identify the most representative structures. The final results included the five top-ranked models, evaluated by energy and clustering metrics, along with an archive of all generated models. The SA profile prediction and cluster data revealed structural diversity among the models [76].

### 3.9. Synthesis of Lead Peptides

The lead peptides were synthesised at S Biochem Company (Thrissur, Kerala) using standard solid-phase peptide synthesis [77]. Identity and purity were verified by reverse-phase HPLC and electrospray ionisation mass spectrometry; representative chromatograms and mass spectra are provided in Appendix A. The peptides were supplied as lyophilised powders, stored at −20 °C and dissolved in sterile water immediately before use.

### 3.10. Safety Assessment of Coconut Milk Peptides by Hemolytic Assay

The hemolytic potential of the lead peptides was determined following an established erythrocyte hemolysis protocol [78]. In brief, 15 mL of venous blood was drawn from a healthy individual, centrifuged at 3000 rpm for 10 min, and the plasma was discarded. The residual pellet (RBCs) was washed 3 times with PBS (pH 7.4, 0.1 M), and the resultant pellet was resuspended in PBS in a 1:10 ratio. Subsequently, 0.9 mL of the RBC suspension was incubated with 0.1 mL of peptides at various concentrations, a negative control (100 μL PBS + 900 μL RBC suspension), and a positive control (100 μL of 0.5% Triton-X-100 + 900 μL of RBC suspension), and incubated at 37 °C for 24 h. Post incubation, all tubes were centrifuged at 3000 rpm for 10 min. 0.1 mL of the supernatant was then transferred to 96-well plates, and the absorbance of the supernatant was read at 540 nm. The percentage inhibition was calculated using Equation (1), and the IC_50_ (HC_50_) of the peptides was determined by analysing the percentage inhibition values in relation to the peptide concentration.(1)$$Inhibition\;\left(\%\right)=\frac{(A_{control}-A_{sample})\;}{A_{control}\;}\cdot 100$$
where $A_{control}$ and $A_{sample}$ denote absorbance at 540 nm corrected for the negative control. The corrected values were calculated as $A_{control}=A_{pos}-A_{neg}$ and $A_{sample}=A_{test}-A_{neg}$ where $A_{pos}$, $A_{neg}$ and $A_{test}$ are the absorbance values of the positive control, negative control, and peptide-treated sample (given concentration), respectively.

### 3.11. α-Glucosidase Inhibition Assay

The α-glucosidase inhibition assay was performed following the method described by Huligere et al. [79]. Briefly, the study involved mixing 0.7 mL of phosphate buffer with 0.1 mL of the test compounds (peptides 1 and peptide 2) at various concentrations, followed by the addition of 0.1 mL of yeast α-glucosidase (0.4 U/mL). In this assay, one unit of activity was defined as the amount of enzyme required to release one µmol of p-nitrophenol from pNPG per minute under the specified experimental conditions. The mixture was pre-incubated for 10 min at 37 °C, and then 0.1 mL of a 0.5 mM pNPG solution in 50 mM phosphate buffer was added. The reaction was maintained at 37 °C for an additional 20 min. Enzyme activity was determined by measuring the absorbance of liberated p-nitrophenol at 405 nm using a microplate reader (Spectramax i3, Molecular Devices, LLC, San Jose, CA, USA). The absorbance was compared with a blank (buffer instead of sample) and acarbose as a positive control, and results were expressed as per cent α-glucosidase inhibition using Equation (1). IC_50_ values were determined by relating the percentage inhibition of each sample to its concentration.

### 3.12. α-Amylase Inhibition Assay

The α-amylase inhibition assay was performed using 1% (*w*/*v*) soluble starch in 0.02 M sodium phosphate buffer (pH 6.9) containing six mM NaCl as the substrate. α-Amylase (0.5 mg/mL) was dissolved in this buffer and preincubated with the test samples (peptides 1 and peptide 2) at various concentrations for 10 min at 25 °C. The enzymatic reaction was initiated by adding the starch substrate and allowed to proceed for 15 min. The reaction was then terminated by adding the 3,5-dinitrosalicylic acid (DNSA) reagent, and the mixture was heated to boiling for 5 min. After appropriate dilution, absorbance was measured at 540 nm using a spectrophotometer (UV-1280 Shimadzu, Duisburg, Germany). Results were compared with a blank control and acarbose, and α-amylase inhibitory activity was expressed as percentage inhibition calculated according to Equation (1). IC_50_ values were obtained from plots of percentage inhibition versus sample concentration [80,81,82].

### 3.13. Aldose-Reductase Inhibition Assay

The aldose reductase inhibition assay was performed according to the method described by Alharbi et al. [83]. Briefly, a 1 mL reaction mixture comprised 0.87 mL of 100 mM Sodium phosphate buffer (pH 6.2), 0.05 mL of 3 mM NADPH, 0.01 mL of human aldose reductase (recombinant, 0.2 U/mL), with the enzyme’s activity being 0.2 IU/mL, and 0.02 mL of test compounds (designated as peptide 1 and peptide 2) at different concentrations. 0.05 mL of the substrate (DL-glyceraldehyde, five mM) was incubated in the reaction mixture for 3 min at 25 °C. Enzyme activity was determined by measuring the reduction in NADPH absorption at 340 nm over 10 min. Inhibitory activity was evaluated by comparing the absorbance change in the test to that in the control (quercetin at various concentrations). Aldose reductase inhibitory activity was expressed as the percentage of inhibition. IC_50_ values were calculated from the analysis of the relationship between the inhibition percentage exhibited by each sample and its respective concentration.

### 3.14. Kinetics of Enzyme Inhibition

Enzyme inhibition by coconut milk peptide 2 was further characterised for α-glucosidase, α-amylase and aldose reductase by steady-state kinetic analysis. Initial reaction velocities (*v*) were measured at several substrate concentrations ([*S*]) in the absence of inhibitor and at three fixed inhibitor concentrations corresponding approximately to 20%, 40% and 60% inhibition (IC_20_, IC_40_ and IC_60_), as determined from the IC_50_ curves described above. In this context, v denotes the initial rate of product formation, whereas [*S*] denotes the concentration of the chromogenic or natural substrate used in each assay. Michaelis explained how v depends on [*S*] for each condition, as described by the Michaelis–Menten equation (Equation (2)):(2)$$v=\frac{V_{max}\left[S\right]}{K_{m}+\left[S\right]}$$
Lineweaver–Burk double-reciprocal plots (Equation (3)) were then constructed by plotting 1/*v* versus 1/[*S*]:(3)$$\frac{1}{v}=\frac{K_{m}}{V_{max}}\cdot \frac{1}{\left[S\right]}+\frac{1}{V_{max}}$$
Michaelis–Menten parameters (*K_m_* and *V_max_*) and apparent inhibition constants (*K_i_*) were obtained from these Lineweaver–Burk plots [84,85,86].

### 3.15. Maillard Reaction Models and Evaluation of Methyl Glyoxal Scavenging Activity

This preliminary assay was conducted to evaluate the methylglyoxal scavenging activity of peptides derived from coconut milk. The Maillard reaction was employed to assess the scavenging potential of peptides against methylglyoxal, using parameters as outlined in previous studies by Risum et al. [50], Zhu et al. [51], and Brings et al. [54], with modifications. The reaction mixture (1 mL) consisted of both peptides at different concentrations, methylglyoxal (0.01 mM), phosphate buffer, and water. It was incubated at two temperatures: room temperature (followed by immediate termination) and 37 °C (for 2 h). Following the incubation time, the reaction was terminated by adding 10 μL of glacial acetic acid. Creatine was used as a positive control. The experiments were performed in triplicate.

The scavenging potential of the peptides against methylglyoxal was determined using the DNPH assay [87]. In brief, 25 μL of the sample was mixed with 975 μL of water and 1000 μL of DNPH reagent. The reaction mixture was shaken well and incubated at 37 °C for 10 min. Following incubation, the reaction was terminated by adding 1000 μL of 1.5 N NaOH, and the absorbance of the yellowish-orange complex was measured at 525 nm using three blanks: the MGO blank and the Creatine blank. Scavenging efficiency was calculated using Equations (4) and (5).(4)$$A_{CORR}=A_{sample}-A_{blank}$$(5)$$MGO_{scavenging}=\left(1-\frac{A_{corr}}{A_{MGO,corr}}\right)\cdot 100$$
where *A_sample_* is the absorbance of the reaction mixture containing methylglyoxal and the test compound (peptide or creatine), *A_blank_* is the absorbance of the reagent blank (without methylglyoxal and without test compound), and *A_MGO,corr_* is the blank-corrected absorbance of the methylglyoxal blank (*A_MGO_*-*A_blank_*). The creatine control was processed in the same way as the peptide samples.

### 3.16. Antioxidant Assays

The antioxidant assays of the test compounds were conducted in vitro, following previous studies, using the standard DPPH, ABTS, and superoxide radical scavenging assays. Regarding the DPPH assay, 0.2 mL of the test samples (peptides 1 and 2) at various concentrations was added to 1.2 mL of a 40 μg/mL DPPH solution in methanol. The mixtures were shaken vigorously and incubated in the dark for 20 min, after which the reduction in DPPH absorption was measured at 517 nm. The samples without DPPH and with methanol served as blanks. Ascorbic acid was used as the positive control. Antioxidant activity was expressed as EC_50_ values, defined as the concentration required to achieve 50% radical scavenging [47].

Concerning the ABTS assay, the ABTS^•+^ radical cation was generated [88,89]. Subsequently, 0.05 mL of the test samples (peptides 1 and 2) at various concentrations was mixed with 4.00 mL of ABTS^•+^ solution and kept in the dark at room temperature for two h. The absorbance was determined at 734 nm using a UV–Vis Spectrophotometer (UV-1280 Shimadzu, Germany). Ascorbic acid was used as the positive control, and samples without the ABTS radical solution served as blanks. The results were expressed as the effective concentration (EC_50_) values.

For assessment of superoxide scavenging, 0.1 M phosphate buffer (pH 7.4) was mixed sequentially with 150 µM nitroblue tetrazolium (NBT), 60 µM phenazine methosulphate, 468 µM NADH and the test peptides (1 and 2) at the various concentrations. After 10 min of incubation at 25 °C in the dark, absorbance was recorded at 560 nm. Samples without NBT served as blanks, and ascorbic acid was used as the positive control. The results were expressed as the effective concentration (EC_50_) values [90].

### 3.17. Statistical Analysis

All experiments were performed in triplicate, and the results are expressed as mean ± standard deviation (SD). Statistical analysis was conducted using one-way analysis of variance (ANOVA), followed by Tukey’s post hoc test to determine significant differences among groups. A *p*-value ≤ 0.05 was considered statistically significant. All analyses were performed using GraphPad Prism software (version 8.0.1).

## 4. Conclusions

This proof-of-concept study shows that coconut milk is an underexplored yet promising source of multifunctional bioactive peptides with potential antihyperglycemic, antiglycation, and antioxidant effects. Using an integrated approach combining peptidomics, in silico analysis, molecular dynamics, and in vitro testing, we identified two short peptides, MQIFVK and ADVFNPR, as promising candidates targeting multiple pathways related to diabetes. Notably, ADVFNPR exhibited broader engagement, including interaction with RAGE, indicating its greater ability to reduce oxidative and carbonyl stress associated with hyperglycemia.

These findings are significant because they show that food-derived peptides can affect multiple pathways associated with type 2 diabetes mellitus, surpassing the limitations of drugs that target only a single aspect. Such peptides could enhance current treatments, mainly when used in combination, for instance, combining peptides with medications like metformin to improve efficacy and reduce side effects.

Producing these peptides via enzymatic hydrolysis offers greater scalability and lower costs than chemical synthesis, making it well-suited for the development pipelines of functional foods and nutraceuticals. As these peptides are food-derived, they might qualify for regulatory pathways such as food or GRAS status instead of conventional drug approval.

Future research should aim to verify their stability, effects in cell and animal models, explore their molecular mechanisms, assess their bioavailability, and develop effective formulations. In summary, this work advances the field of natural-product-based diabetes treatment and highlights coconut milk-derived peptides as promising candidates for next-generation functional foods and therapeutic uses.

## Figures and Tables

**Figure 1 ijms-27-00360-f001:**
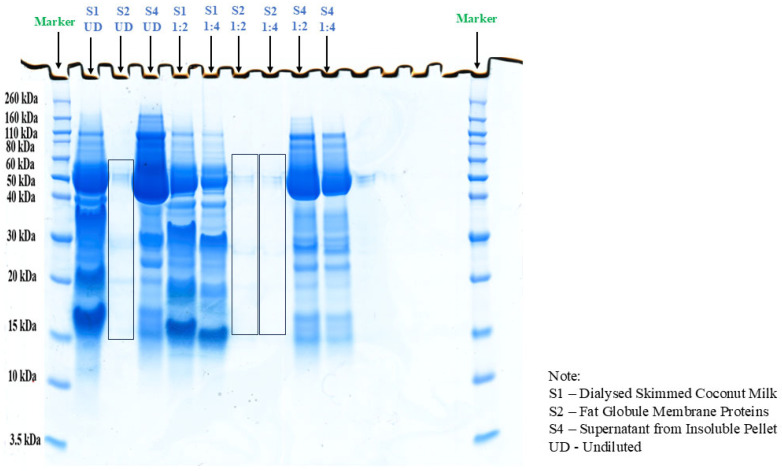
SDS-PAGE profile of proteins isolated from coconut milk.

**Figure 2 ijms-27-00360-f002:**
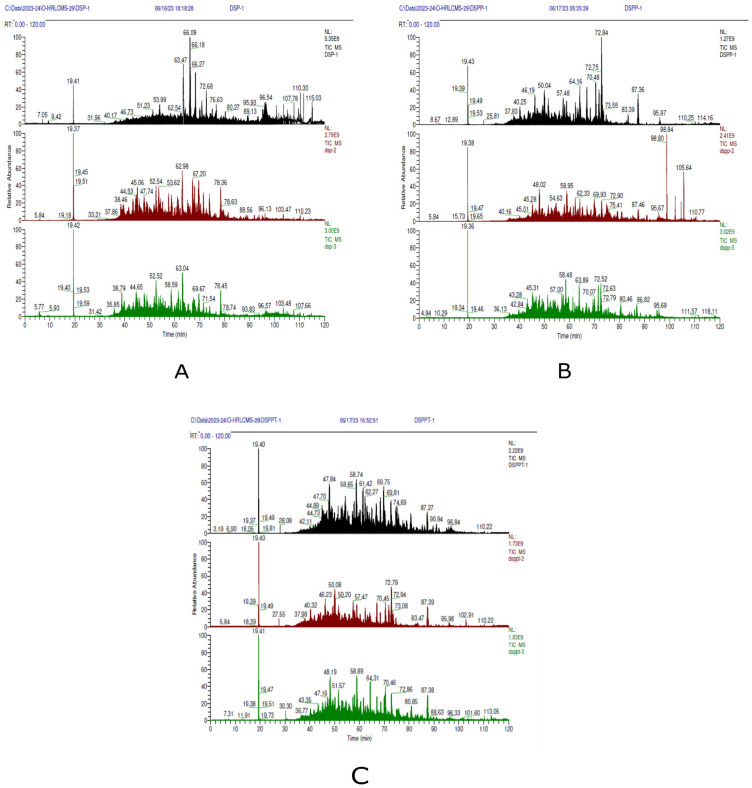
Chromatogram of coconut milk protein hydrolysates: (**A**) Chromatogram of coconut milk protein hydrolysates obtained from pepsin-digestion (DSP-1, 2 and 3 indicate desalted protein hydrolysates obtained from pepsin digestion, 1—Dialysed skimmed coconut milk, 2—Fat globule membrane suspension and 3—supernatant from insoluble pellet). (**B**) Chromatogram of coconut milk protein hydrolysates obtained from pepsin and pancreatin digestion (DSPP-1, 2 and 3 indicate desalted protein hydrolysates obtained from pepsin and pancreatin digestion) and (**C**) Chromatogram of coconut milk protein hydrolysates obtained from pepsin, pancreatin, and trypsin digestion (DSPPT indicate desalted protein hydrolysates obtained from pepsin, pancreatin, and trypsin digestion).

**Figure 3 ijms-27-00360-f003:**
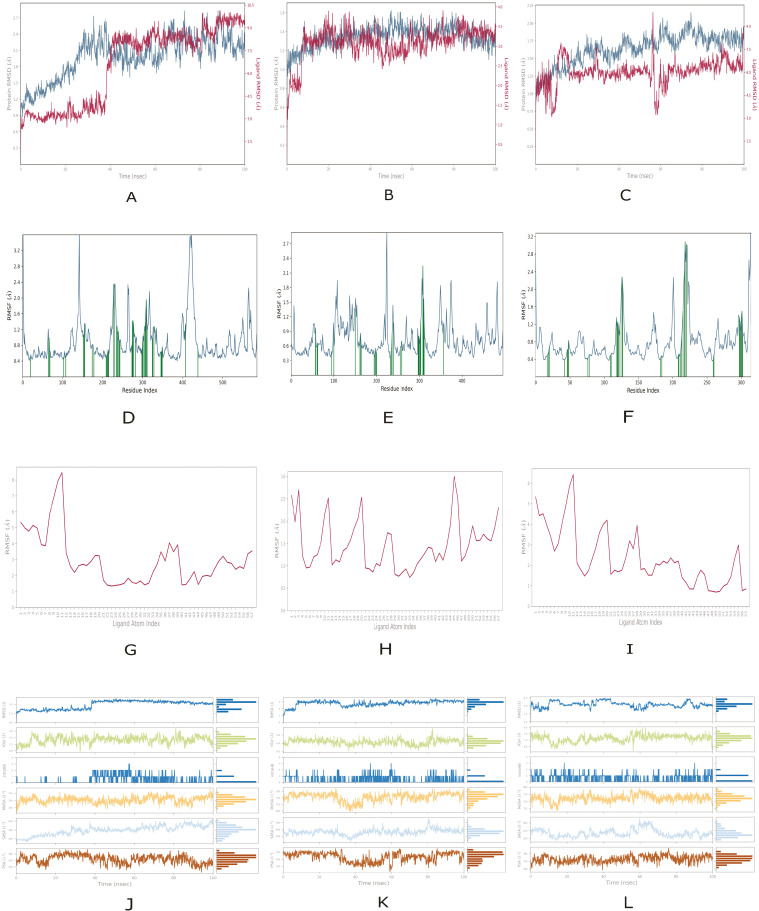
Molecular dynamics trajectories for peptide 1 in complex with diabetic enzyme targets: (**A**–**C**) Protein backbone root mean square deviation (RMSD) for α-glucosidase, α-amylase and aldose reductase in complex with peptide 1, respectively. (**D**–**F**) Per-residue protein root mean square fluctuations (RMSF) for α-glucosidase, α-amylase and aldose reductase, respectively. (**G**–**I**) Ligand RMSF for peptide 1 in the α-glucosidase, α-amylase and aldose reductase binding pockets, respectively. (**J**–**L**) Time evolution of ligand structural descriptors for peptide 1 (RMSD, radius of gyration, molecular surface area (MolSA), solvent accessible surface area (SASA), polar surface area (PSA) and intramolecular hydrogen bonds) in complex with α-glucosidase, α-amylase and aldose reductase, respectively.

**Figure 4 ijms-27-00360-f004:**
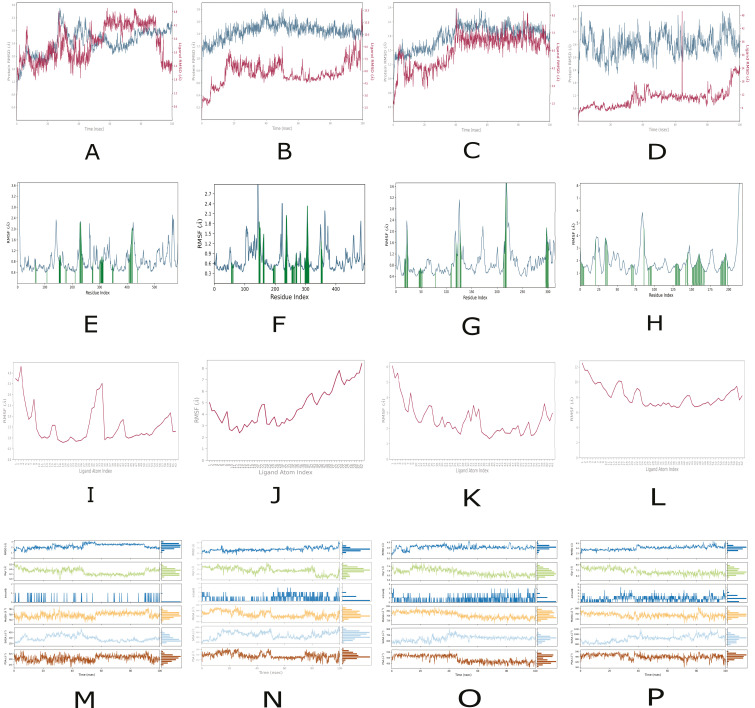
Molecular dynamics trajectories for peptide 2 in complex with diabetic enzyme targets and RAGE: (**A**–**D**) Protein backbone RMSD for α-glucosidase, α-amylase, aldose reductase and RAGE in complex with peptide 2, respectively. (**E**–**H**) Per-residue protein RMSF for α-glucosidase, α-amylase, aldose reductase and RAGE, respectively. (**I**–**L**) Ligand RMSF for peptide 2 in the α-glucosidase, α-amylase, aldose reductase and RAGE binding pockets, respectively. (**M**–**P**) Time evolution of ligand structural descriptors for peptide 2 (RMSD, radius of gyration, MolSA, SASA, PSA and intramolecular hydrogen bonds) in complex with α-glucosidase, α-amylase, aldose reductase and RAGE, respectively.

**Figure 5 ijms-27-00360-f005:**
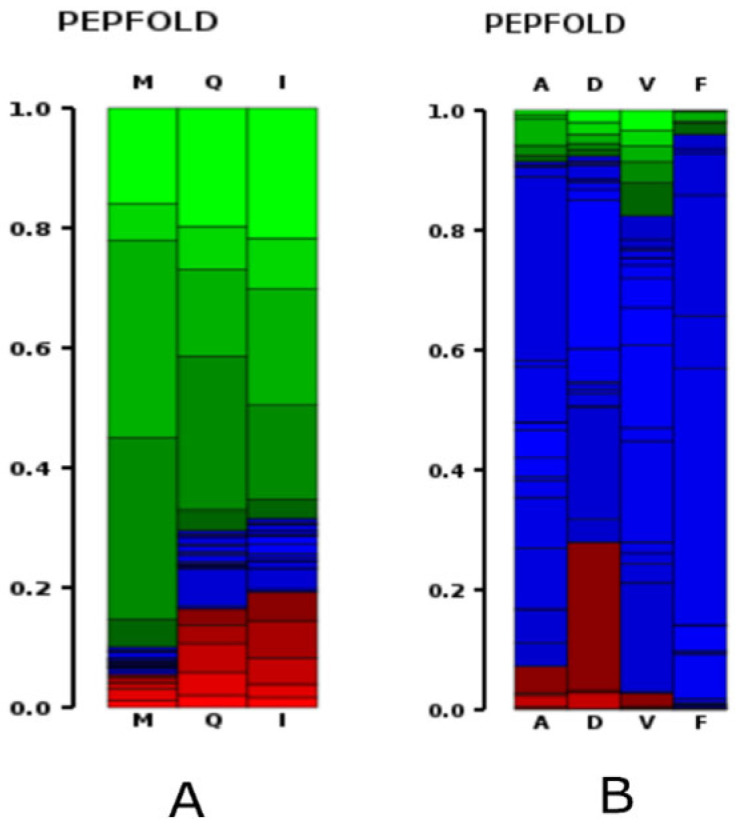
PEP-FOLD maps of secondary-structure propensity: (**A**) Predicted secondary-structure probabilities for peptide 1 (MQIFVK). (**B**) Predicted secondary-structure probabilities for peptide 2 (ADVFNPR).

**Figure 6 ijms-27-00360-f006:**
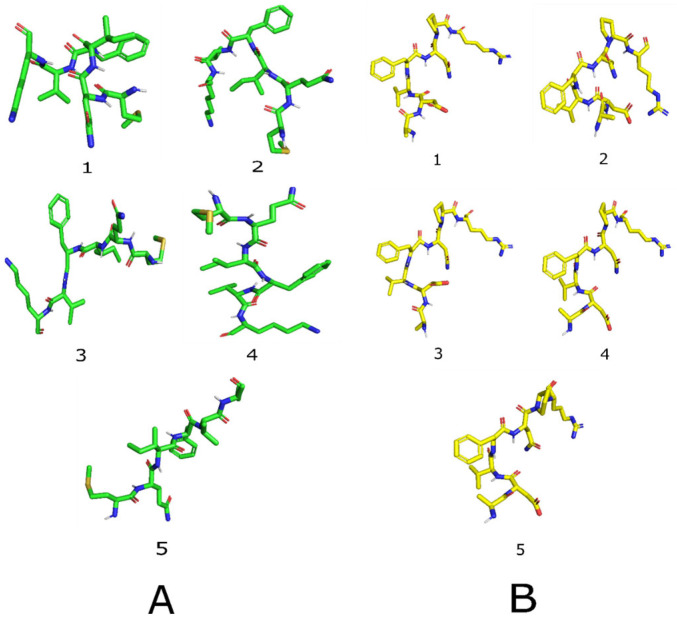
Top-ranked three-dimensional models of the lead peptides predicted by PEP-FOLD: (**A**) Five lowest-energy models of peptide 1 (MQIFVK) are shown in green. (**B**) Five lowest-energy models of peptide 2 (ADVFNPR) are shown in yellow.

**Table 1 ijms-27-00360-t001:** Protein content in different layers of coconut milk after centrifugation. Concentration of protein is calculated with the density of coconut milk as 1.01 g/mL.

Samples	Concentration of Protein (µg/mL)	Concentration of Protein (g/100 g)
Skimmed coconut milk after dialysis	369	0.0365
Crude fat globule membrane (FGM) suspension	109	0.0108
Supernatant from insoluble pellet	1574	0.1558

**Table 2 ijms-27-00360-t002:** Outcomes of molecular docking of coconut milk bioactive peptides.

Peptide/ Standard Drug	Anti-Diabetic Targets						Receptor for Advanced Glycation End-Products (RAGE)	
	α-Glucosidase		α-Amylase		Aldose Reductase			
	Glide Score (kcal/mol)	THB	Glide Score (kcal/mol)	THB	Glide Score (kcal/mol)	THB	Glide Score (kcal/mol)	THB
Peptide 1	−9.34	3	−9.64	4	−11.42	4	-	-
Peptide 2	−9.87	3	−9.87	6	−9.54	3	−8.75	6
* Acarbose	−12.33	7	−11.40	6	-	-	-	-
^#^ Quercetin	-	-	-	-	−10.767	3	-	-
^&^ Papaverine	-	-	-	-	-	-	−3.74	1

* Acarbose is used as a positive control against α-glucosidase and α-amylase, ^#^ Quercetin is used as a positive control against aldose reductase, ^&^ Papaverine is used as a positive control against RAGE and THB—total hydrogen bonds.

## Data Availability

The original contributions presented in this study are included in the article/Appendix A. Further inquiries can be directed to the corresponding author.

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
