# Peer review of "Coconut Milk-Derived Bioactive Peptides as Multifunctional Agents Against Hyperglycemia, Oxidative Stress, and Glycation: An Integrated Experimental and Computational Study"

_ijms, 2025, doi:10.3390/ijms27010360_

Round 1
Reviewer 1 Report
Comments and Suggestions for Authors
The study addresses an interesting topic concerning coconut milk–derived peptides and their potential multifunctional anti-diabetic activities. However, major revisions are required before the manuscript can be considered for publication. My comments are as follows:
- The introduction does not align with the core focus of the study.
The title clearly indicates that the topic of the work is the discovery, characterization, and biological relevance of bioactive peptides from coconut milk. However, the Introduction has an excessive number of paragraphs to describe diabetes pathophysiology, including detailed descriptions of hyperglycemia, AGEs, RAGE signalling, oxidative stress, and related molecular pathways. In contrast, the sections discussing the state of research on coconut proteins, coconut-derived peptides, food-derived peptides, and the rationale for selecting coconut milk as a source are comparatively limited. The authors should (i) streamline the biomedical background, (ii) expand the discussion on coconut milk protein composition and previous work on food-derived peptides, and (iii) clearly explain why coconut milk proteins are worth investigating and why the three enzymatic targets were chosen.
- The Results are highly descriptive but lack critical analysis and interpretation. Several subsections present excessive detail while missing essential insights:
Section 2.2: The manuscript provides numerous peptide counts from different food matrices, but does not explain why the identified coconut peptides are important or how their characteristics support their potential bioactivity.
Section 2.4: The MD simulation results include detailed description of RMSD/RMSF fluctuations over time but do not address the major findings. The authors must answer essential questions such as: 1) Which peptide is more stable overall? 2) Which target complexes are most stable and why? 3) How does MD support or validate the docking and in vitro results? 4) What is the biological implication of phenomena such as RMSD > 24 Å for peptide 2 in the RAGE simulation?
Section 2.5: The authors describe sOPEP scores and Ramachandran-like plots in unnecessary detail but fail to explain why structural stability matters for multi-target binding and how the predicted conformations support the peptides’ biological activities.
Overall, the Results section should emphasize interpretation, not simply present descriptive results.
- Repetition of concepts and statements should be avoided.
Several parts of the manuscript repeat definitions or statements unnecessarily:
The meaning of HCâ‚…â‚€ is explained multiple times in Section 2.6.
In Section 2.7, the sentence “Among the test compounds, peptide 2 showed the highest inhibition” (or an equivalent phrasing) appears in two consecutive paragraphs (approximately at lines 564 and 591).
The definition of BCA also appears in two separate locations (lines 807 and 121 respetively).
- Literature comparison is excessive and distracts from the main narrative.
Although comparing the findings with existing literature is valuable, the manuscript frequently provides overly long lists of citations and detailed numerical comparisons of ICâ‚…â‚€ or ECâ‚…â‚€ values from numerous studies. In several sections (e.g., 2.7 and 2.8), the authors cite 5–10 papers in a row, report their exact values, and provide extended commentary. This leads to the Results and Discussion sections reading like a mini-review rather than a focused scientific analysis of new experimental findings.
The authors should condense these comparisons and highlight only the most relevant studies. Excessive literature descriptions should be moved to Discussion or removed entirely.
- The Conclusion is overly long and lacks a clear high-level message.
The conclusion currently lists seven separate methodological suggestions for future work, which reads more like a procedural checklist than a scientific perspective. A conclusion should be concise, emphasize the scientific significance of the study, and present broad, directional future research recommendations.
Additionally, the authors should explicitly state why the findings matter to the scientific community. For example, how these peptides contribute to natural-product-based antidiabetic research, food-derived therapeutics, or functional nutrition.
Author Response
Reviewer 01
The study addresses an interesting topic concerning coconut milk–derived peptides and their potential multifunctional anti-diabetic activities. However, major revisions are required before the manuscript can be considered for publication. My comments are as follows:
- The introduction does not align with the core focus of the study.
The title clearly indicates that the topic of the work is the discovery, characterization, and biological relevance of bioactive peptides from coconut milk. However, the Introduction has an excessive number of paragraphs to describe diabetes pathophysiology, including detailed descriptions of hyperglycemia, AGEs, RAGE signalling, oxidative stress, and related molecular pathways. In contrast, the sections discussing the state of research on coconut proteins, coconut-derived peptides, food-derived peptides, and the rationale for selecting coconut milk as a source are comparatively limited. The authors should (i) streamline the biomedical background, (ii) expand the discussion on coconut milk protein composition and previous work on food-derived peptides, and (iii) clearly explain why coconut milk proteins are worth investigating and why the three enzymatic targets were chosen.
Authors' response: Firstly, we thank the reviewer for their valuable suggestions and concerns regarding the manuscript quality. As per the suggestion, the introduction has been modified to emphasize the composition of coconut milk proteins and clearly explain why coconut milk proteins are worth exploring, as well as the rationale for selecting three enzymatic targets.
- The Results are highly descriptive but lack critical analysis and interpretation. Several subsections present excessive detail while missing essential insights:
Section 2.2: The manuscript provides numerous peptide counts from different food matrices, but does not explain why the identified coconut peptides are important or how their characteristics support their potential bioactivity.
Author’s response: We thank the reviewer for their valuable suggestion. As per the suggestion, section 2.2 has been modified to emphasize the importance of the identified coconut peptides and their characteristics, which support their potential bioactivity.
Section 2.4: The MD simulation results include detailed description of RMSD/RMSF fluctuations over time but do not address the major findings. The authors must answer essential questions such as: 1) Which peptide is more stable overall? 2) Which target complexes are most stable and why? 3) How does MD support or validate the docking and in vitro results? 4) What is the biological implication of phenomena such as RMSD > 24 Å for peptide 2 in the RAGE simulation?
Author’s response: We thank the reviewer for their valuable suggestion. The responses for the queries have been separately provided below
Peptide 1 is more stable overall compared to peptide 2 and the explanation for this has been incorporated in the manuscript.
Among the enzyme targets, the peptide 1–α-amylase complex stands out as the most dynamically stable system. It shows low protein RMSD, limited residue fluctuations, and moderate ligand RMSD, suggesting effective peptide anchoring in the catalytic cleft. This stability aligns with the deep, well-defined active site of α-amylase, which promotes stable peptide binding and prevents excessive ligand movement. The same explanation has been included in the manuscript.
MD simulations enhance static docking predictions by revealing which peptide–target interactions are maintained over time. Peptide 1, which shows potent inhibitory effects in vitro, consistently demonstrates longer binding durations and less ligand movement in MD simulations, especially with α-amylase and α-glucosidase. Conversely, peptide 2 tends to have greater conformational flexibility and lower retention, notably in the α-amylase and RAGE systems. However, peptide 2 showed more activity than peptide 1 in vitro. This link between dynamic stability and experimental activity supports using MD as a mechanistic validation tool that connects docking results with biological effects. Significantly, ligand rearrangements during MD do not destabilize the protein, suggesting that inhibition occurs through reversible, adaptive binding modes compatible with physiological conditions.
Section 2.5: The authors describe sOPEP scores and Ramachandran-like plots in unnecessary detail but fail to explain why structural stability matters for multi-target binding and how the predicted conformations support the peptides’ biological activities.
Overall, the Results section should emphasize interpretation, not simply present descriptive results.
Author’s response: As per the reviewer’s suggestion, the results & discussion have been modified and included in the manuscript, explaining the relationship between structural stability in multi-target binding and how the predicted conformations support the peptides’ biological activities. As this is a preliminary study, we have focused on the SOPEP score, model conformations, and initial analysis. However, with due respect to the reviewer’s query, we will consider this point and conduct detailed structural and molecular analysis of the peptides in future studies.
- Repetition of concepts and statements should be avoided.
Several parts of the manuscript repeat definitions or statements unnecessarily:
The meaning of HCâ‚…â‚€ is explained multiple times in Section 2.6.
Author’s response: As per the reviewer’s suggestion, repetition of the meaning of HCâ‚…â‚€ has been removed, and the safety assessment section has been modified. However, we confirm that the meaning has been retained.
In Section 2.7, the sentence “Among the test compounds, peptide 2 showed the highest inhibition” (or an equivalent phrasing) appears in two consecutive paragraphs (approximately at lines 564 and 591).
Authors response: As per the reviewer’s suggestion, repetition of the sentence “Among the test compounds, peptide 2 showed the highest inhibition has been removed, and the meaning of the entire section is retained. We also checked the manuscript carefully and corrected other minor editing and language shortcomings
The definition of BCA also appears in two separate locations (lines 807 and 121 respetively).
Author’s response: As per the reviewer’s suggestion, the repeated definition of BCA has been removed.
- Literature comparison is excessive and distracts from the main narrative.
Although comparing the findings with existing literature is valuable, the manuscript frequently provides overly long lists of citations and detailed numerical comparisons of ICâ‚…â‚€ or ECâ‚…â‚€ values from numerous studies. In several sections (e.g., 2.7 and 2.8), the authors cite 5–10 papers in a row, report their exact values, and provide extended commentary. This leads to the Results and Discussion sections reading like a mini-review rather than a focused scientific analysis of new experimental findings.
The authors should condense these comparisons and highlight only the most relevant studies. Excessive literature descriptions should be moved to Discussion or removed entirely.
Author’s response: We thank the reviewer for their valuable suggestion. As per the suggestion, the comparative results have been condensed, with only the most relevant studies highlighted. Excessive descriptions of the literature have been removed to avoid confusion.
- The Conclusion is overly long and lacks a clear high-level message.
The conclusion currently lists seven separate methodological suggestions for future work, which reads more like a procedural checklist than a scientific perspective. A conclusion should be concise, emphasize the scientific significance of the study, and present broad, directional future research recommendations.
Additionally, the authors should explicitly state why the findings matter to the scientific community. For example, how these peptides contribute to natural-product-based antidiabetic research, food-derived therapeutics, or functional nutrition.
Author’s response: We thank the reviewer for their valuable suggestion; the conclusion section has been modified as per the reviewer’s suggestions. The current conclusion is concise, emphasizes the study's scientific significance, and presents broad, directional future research recommendations. We have also stated why the findings matter to the scientific community.
Overall, we hope these revisions address the reviewers' concerns to the best of our knowledge.
Reviewer 2 Report
Comments and Suggestions for Authors| Clarifications | • Define FGM early (fat globule membrane). • Clarify protein concentration discrepancy. • Add µM to all ICâ‚…â‚€/ECâ‚…â‚€ tables. |
| Supplementary | • Include HPLC purity chromatograms (S3/S4 already mentioned—ensure data present). • Provide MM/GBSA ΔG_bind for MD complexes. |
| Discussion Expansion | • Address RAGE binding instability (allostery? cryptic pocket?). • Discuss bioavailability challenges (oral stability, permeability). • Compare ADVFNPR with known RAGE inhibitors (e.g., Azeliragon, FPS-ZM1). |
| Figures/Tables | • Merge Tables 3–7 into a single “Bioactivity Profile” table for quick comparison. • Annotate key interacting residues in Fig. S1 (2D diagrams). |
| Ethics/Transparency | • State donor health status for RBCs (healthy vs. diabetic). • Disclose peptide synthesis purity certificate (if available). |
Introduction: Could better emphasize why coconut milk is uniquely promising: e.g., high Arg content (cited in refs [14,27]) supports MGO scavenging—but this is only brought up later in Results. Suggest moving this insight earlier.
Results and discussion: Table 1: protein content in skimmed milk (0.0365 g/100g) seems low vs. literature (~0.2–0.5 g/100g). Clarify if this reflects dialysis loss or dilution during extraction. Also, “Crude FGM suspension” abbreviation not defined earlier. Table 2: Glide scores for acarbose unusually low (−12.33 for α-glucosidase)—in most literature, experimentally bound acarbose scores ~−8 to −10. Verify docking protocol (e.g., grid constraints, ligand prep). Possible overestimation may inflate perceived superiority. RAGE–peptide 2 complex shows extreme ligand RMSD (>24 Å). While authors acknowledge this, functional implications need deeper discussion: is RAGE inhibition allosteric? Or is docking pose inaccurate? Consider MM/GBSA binding energy calculation to complement RMSD. No validation of predicted folds; suggest adding: “Future CD/NMR studies will confirm these conformational preferences." Human erythrocytes used, but donor health status (e.g., diabetic vs. normoglycemic) undisclosed. MGO/RAGE axis dysregulation in diabetes may alter membrane susceptibility.Missing: Purity of synthesized peptides (stated >95% in text, but no HPLC trace shown in Suppl.). Enzyme sources/specifications (e.g., α-glucosidase from S. cerevisiae—commercial grade? Specific activity?). Statistical method for ICâ‚…â‚€/ECâ‚…â‚€ curve fitting (e.g., log[inhibitor] vs. response—4PL?)
Conclusion: Underemphasized:
Potential synergy with existing drugs (e.g., ADVFNPR + metformin).
Scalability of peptide production (enzymatic hydrolysis vs. synthesis cost).
Regulatory pathway (GRAS? peptide drug vs. nutraceutical?).
Author Response
Reviewer 02
|
Clarifications |
• Define FGM early (fat globule membrane). • Clarify protein concentration discrepancy. • Add µM to all ICâ‚…â‚€/ECâ‚…â‚€ tables. |
|
Supplementary |
• Include HPLC purity chromatograms (S3/S4 already mentioned—ensure data present). • Provide MM/GBSA ΔG bind for MD complexes. |
|
Discussion Expansion |
• Address RAGE binding instability (allostery? cryptic pocket?). • Discuss bioavailability challenges (oral stability, permeability). • Compare ADVFNPR with known RAGE inhibitors (e.g., Azeliragon, FPS-ZM1). |
|
Figures/Tables |
• Merge Tables 3–7 into a single “Bioactivity Profile” table for quick comparison. • Annotate key interacting residues in Fig. S1 (2D diagrams). |
|
Ethics/Transparency |
• State donor health status for RBCs (healthy vs. diabetic). • Disclose peptide synthesis purity certificate (if available). |
1) Clarifications
(a) Define FGM early (fat globule membrane)
Author’s response: As per the reviewer's suggestions, FGM has been defined early as the fat globule membrane.
(b) Clarify protein concentration discrepancy
Author’s response: Firstly, we would like to confirm that the protein extraction followed scientific protocols. However, the discrepancy in protein concentration could be due to dilution and dialysis. As this is a preliminary study, we will definitely validate in our future studies.
(c) Add µM to all ICâ‚…â‚€/ECâ‚…â‚€ tables.
Author’s response: All the IC50/EC50 values have also been represented in µM (the values are given in brackets below each value in the combined Table 4) in addition to the µg/mL.
2) Supplementary - As per the reviewer’s suggestions,
(a) Include HPLC purity chromatograms (S3/S4 already mentioned—ensure data present).
Author’s response: We ensure that data related to purity of peptides (in the form of HPLC chromatograms i.e. Supplementary figures S3 and S4) is present.
(b) Provide MM/GBSA ΔG bind for MD complexes.
Author’s response: We acknowledge that we have not performed MM-GBSA/ΔG binding calculations for MD complexes. The results presented in the manuscript are based solely on the trajectories. However, we definitely consider this a valuable suggestion and perform MM-GBSA/ΔG binding for MD complexes in the upcoming studies. Hence, we hope these revisions address the reviewer’s concern.
3) Discussion Expansion
(a) Address RAGE binding instability (allostery? cryptic pocket?).
Author’s response: We thank the reviewer for highlighting the need for a more in-depth discussion of the peptide 2–RAGE interaction. We have included the explanation in the manuscript to clarify that the large ligand RMSD (>24 Å) observed in the RAGE simulation reflects biologically relevant weak or transient binding rather than a simulation artefact. The peptide remains internally compact, indicating translational mobility on the RAGE surface rather than unfolding. This behaviour is consistent with the known structural features of RAGE, which binds diverse ligands via shallow, flexible surface interfaces rather than deep binding pockets. The observed instability in the peptide 2–RAGE complex likely reflects surface-mediated or atypical interactions rather than stable allosteric inhibition. While RMSD analysis alone cannot conclusively identify allosteric modulation, the absence of a consistent binding configuration suggests that peptide 2 is unlikely to serve as a potent RAGE inhibitor. However, it may transiently interact with the receptor (could be a modulator). While MM/GBSA binding free-energy calculations could provide additional thermodynamic insight, these were beyond the scope of the present study and will be considered in future work. We have also acknowledged the need to validate peptide conformations using CD or NMR spectroscopy experimentally. A detailed discussion has been included in the manuscript with relevant references to the best of our knowledge. Hence, we request the reviewer to accept the manuscript.
(b) Discuss bioavailability challenges (oral stability, permeability).
Author’s response: We thank the reviewer for the valuable suggestion. As per the reviewer’s suggestion, the bioavailability challenges have been discussed in the manuscript. However, we acknowledge that bioavailability studies (including oral stability and permeability) were not conducted in the current study. But we definitely consider this point and perform them in future studies.
(c) Compare ADVFNPR with known RAGE inhibitors (e.g., Azeliragon, FPS-ZM1).
Author’s response: We thank the reviewer for their valuable suggestion to include a comparative explanation with known RAGE inhibitors. As per the reviewer’s suggestion, the comparative explanation has been provided in the results and discussion section of the molecular dynamics simulation section to the best of our knowledge. Hence, we request the reviewer to accept the manuscript.
4) Figures/Tables
(a) Merge Tables 3–7 into a single “Bioactivity Profile” table for quick comparison.
Author’s response: We thank the reviewer for their valuable suggestion. As per the reviewer’s suggestion, Tables 3, 4, 6 & 7 have been merged as Table 4. However, due to limited space, Table 5 could not be merged. Hence, we request that the author consider the current table merge and accept the manuscript. Additionally, we ensure that the merged table (Table 4) has been cited appropriately.
(b) Annotate key interacting residues in Fig. S1 (2D diagrams).
Author’s response: We thank the reviewer for their valuable suggestion. As per the reviewer’s suggestion, key interacting residues have been annotated with a green box in Fig. S1 (2D diagrams).
- Ethics/Transparency
(a) State donor health status for RBCs (healthy vs. diabetic).
Author’s response: We thank the reviewer for their valuable suggestion. We want to confirm that the blood donor was healthy. As this is a preliminary study, we aimed solely to assess the hemolytic potential (safety) of the synthesised peptides. However, with due respect to the reviewer’s suggestion, we will consider it and conduct a comparative study in our future work. The donor's health status is described in the safety assessment methodology section.
(b) Disclose peptide synthesis purity certificate (if available).
Author’s response: We thank the reviewer for their valuable query. In our study, the identified peptides were synthesised by S Biochem, a well-known, certified company that synthesizes and conducts research on peptides. As per the protocol, they have shared the chromatogram and mass spectrogram of the peptides (included in the supplementary material), which indicates their purity. We want to confirm that no purity certificate is available. Hence, we request the reviewer to accept the manuscript.
Introduction: Could better emphasize why coconut milk is uniquely promising: e.g., high Arg content (cited in refs [14,27]) supports MGO scavenging—but this is only brought up later in Results. Suggest moving this insight earlier.
Author’s response: As per the reviewer’s suggestion, we have modified the introduction to emphasize the key points of coconut milk with respect to the bioactivities.
Results and discussion: Table 1: protein content in skimmed milk (0.0365 g/100g) seems low vs. literature (~0.2–0.5 g/100g). Clarify if this reflects dialysis loss or dilution during extraction. Also, “Crude FGM suspension” abbreviation not defined earlier. Table 2: Glide scores for acarbose unusually low (−12.33 for α-glucosidase)—in most literature, experimentally bound acarbose scores ~−8 to −10. Verify docking protocol (e.g., grid constraints, ligand prep). Possible overestimation may inflate perceived superiority. RAGE–peptide 2 complex shows extreme ligand RMSD (>24 Å). While authors acknowledge this, functional implications need deeper discussion: is RAGE inhibition allosteric? Or is docking pose inaccurate? Consider MM/GBSA binding energy calculation to complement RMSD. No validation of predicted folds; suggest adding: “Future CD/NMR studies will confirm these conformational preferences." Human erythrocytes used, but donor health status (e.g., diabetic vs. normoglycemic) undisclosed. MGO/RAGE axis dysregulation in diabetes may alter membrane susceptibility.Missing: Purity of synthesized peptides (stated >95% in text, but no HPLC trace shown in Suppl.). Enzyme sources/specifications (e.g., α-glucosidase from S. cerevisiae—commercial grade? Specific activity?). Statistical method for ICâ‚…â‚€/ECâ‚…â‚€ curve fitting (e.g., log[inhibitor] vs. response—4PL?)
- Table 1: protein content in skimmed milk (0.0365 g/100g) seems low vs. literature (~0.2–0.5 g/100g). Clarify if this reflects dialysis loss or dilution during extraction. Also, “Crude FGM suspension” abbreviation not defined earlier.
Author’s response: Firstly, we would like to confirm that the protein extraction followed scientific protocols. However, the protein concentration in skimmed milk (0.0365 g/100g) appears low compared to the literature (~0.2–0.5 g/100g), which may be due to dialysis loss. Also, the abbreviation of crude FGM suspension has been defined.
- Table 2: Glide scores for acarbose unusually low (−12.33 for α-glucosidase)—in most literature, experimentally bound acarbose scores ~−8 to −10. Verify docking protocol (e.g., grid constraints, ligand prep). Possible overestimation may inflate perceived superiority.
Author’s response: Firstly, we want to thank the reviewer for their keen observation. We also accept that, in most of the literature, experimentally bound acarbose scores are ~−8 to −10. We want to clarify that docking was performed using the Glide XP protocol, with the exact grid definition and ligand preparation for all compounds to ensure consistent comparisons. The lower (more negative) score for acarbose indicates its high flexibility and ability to form extensive hydrogen bonds, which are known to yield lower Glide XP scores, depending on ligand setup and scoring methods. It is important to note that docking was not the sole criterion for qualitative ranking, nor for precise affinity measurement, and no claim of superiority over acarbose was made. Our main conclusions are based on the experimental inhibition data. Hence, we request the reviewer to accept the manuscript.
(c) RAGE–peptide 2 complex shows extreme ligand RMSD (>24 Å). While authors acknowledge this, functional implications need deeper discussion: is RAGE inhibition allosteric? Or is docking pose inaccurate? Consider MM/GBSA binding energy calculation to complement RMSD. No validation of predicted folds; suggest adding: “Future CD/NMR studies will confirm these conformational preferences."
Author’s response: We thank the reviewer for highlighting the need for deeper discussion of the peptide 2–RAGE interaction. We have expanded the manuscript to clarify that the large ligand RMSD (>24 Å) observed in the RAGE simulation reflects biologically relevant weak or transient binding rather than a simulation artefact. The peptide remains internally compact, indicating translational mobility on the RAGE surface rather than unfolding. This behaviour is consistent with the known structural features of RAGE, which binds diverse ligands via shallow, flexible surface interfaces rather than deep binding pockets. We have avoided overinterpreting this behaviour as allosteric inhibition and instead discuss it as a non-classical or surface-mediated interaction. While MM/GBSA binding free-energy calculations could provide additional thermodynamic insight, these were beyond the scope of the present study and will be considered in future work. We have also acknowledged the need to validate peptide conformations using CD or NMR spectroscopy experimentally. A detailed discussion has been included in the manuscript with relevant references to the best of our knowledge. Hence, we request the reviewer to accept the manuscript.
(d) Human erythrocytes used, but donor health status (e.g., diabetic vs. normoglycemic) undisclosed.
Author’s response: We thank the reviewer for their valuable suggestion. We want to confirm that the blood donor was healthy. As this is a preliminary study, we aimed solely to assess the hemolytic potential (safety) of the synthesised peptides. However, with due respect to the reviewer’s suggestion, we will consider it and conduct a comparative study in our future work. The donor's health status is described in the safety assessment methodology section.
(e) MGO/RAGE axis dysregulation in diabetes may alter membrane susceptibility.
Author’s response: We thank the reviewer for their keen observation and the valuable query. Chronic hyperglycaemia in diabetes is known to elevate methylglyoxal levels and activate the AGE–RAGE axis, which promotes oxidative stress, lipid peroxidation, and non-enzymatic glycation of membrane proteins, thereby increasing membrane fragility and susceptibility to damage. In the present study, hemolytic activity was evaluated using erythrocytes from healthy donors as a standard first-line biocompatibility assessment. However, we acknowledge that diabetic membranes may exhibit altered susceptibility due to MGO/RAGE-mediated dysregulation. In this study, we did not perform a susceptibility study, and we would definitely consider this point for our future work to evaluate peptide safety under glycation- or diabetes-mimicking conditions. Hence, we request the reviewer to accept the manuscript.
(f) Missing: Purity of synthesized peptides (stated >95% in text, but no HPLC trace shown in Suppl.).
Author’s response: We thank the reviewer for their valuable query. In our study, the identified peptides were synthesised by S Biochem, a well-known, certified company that synthesizes and conducts research on peptides. As per the protocol, they have shared the chromatogram and mass spectrogram of the peptides (included in the supplementary material), which indicates their purity. We want to confirm that no purity certificate is available and that it has not been missed. Hence, we request the reviewer to accept the manuscript.
(g) Enzyme sources/specifications (e.g., α-glucosidase from S. cerevisiae—commercial grade? Specific activity?).
Author’s Response: As per the reviewer’s suggestion, enzyme sources and specifications have been provided in the manuscript in the materials section. All enzymes used were commercial-grade and purchased from Sisco Research Laboratories (SRL).
(h) Statistical method for ICâ‚…â‚€/ECâ‚…â‚€ curve fitting (e.g., log[inhibitor] vs. response—4PL?)
Author’s Response: As per the reviewer’s query, we used log[inhibitor] vs. response—4PL to calculate IC50 values from the percentage inhibition vs. concentration data using GraphPad Prism (v 8.0.1).
Conclusion: Underemphasized:
Potential synergy with existing drugs (e.g., ADVFNPR + metformin).
Scalability of peptide production (enzymatic hydrolysis vs. synthesis cost).
Regulatory pathway (GRAS? peptide drug vs. nutraceutical?)
Authors' response: We thank the reviewer for their valuable suggestion. The conclusion section has been modified by emphasizing the points suggested by the reviewer.
Overall, we hope these revisions address the reviewers' concerns to the best of our knowledge.
Reviewer 3 Report
Comments and Suggestions for Authors
This work titled "Coconut milk-derived bioactive peptides as multifunctional agents against hyperglycemia, oxidative stress, and glycation: An integrated experimental and computational study" mostly shows an alternative research focusing on natural resources in the quest for a therapy that integrates anti-hyperglycemic, antioxidant, and anti-glycation properties, along with potent inhibition of α-glucosidase, α-amylase, and aldose reductase, as well as suppression of AGEs through RAGE, can be optimal in controlling diabetes and its complications. Authors based on the literature information about how chemical analyses indicate that CM is rich in nutritionally and biologically active constituents, such as phenolic derivatives, a variety of fatty acids, antioxidant molecules, simple sugars, complex carbohydrates and structural proteins [13]. From there, authors developed this work aiming to study the anti-hyperglycemic and anti-glycation properties of coconut milk bioactive peptides using in silico and in vitro approaches, which will aid future research in developing an alternative food-derived bioactive molecule to reduce glucose and, consequently, advanced glycation end products.
According to the authors, the protein analysis show results are in line with the previous study by Kwon et al. [16], which identified five different proteins: albumins, globulins, prolamines, glutelins-1, and glutelins-2.The predominant protein in coconut (65%) is 11S globulin, known as Cocosin [19]. Cocosin is thought to be more crucial in maintaining the stability of coconut milk compared to albumin or the 7S globulin fraction [18]. After protein extraction and SDS-PAGE, peptides were identified using nano-ESI-Orbitrap-LC-MS/MS, and their sequences were compared with the Cocos nucifera proteome library. A total of 114 peptide sequences have been identified, of which 96 unique peptides were retained after removal of duplicates - were predicted to be stable, with an instability index below 40. Seventeen of these stable peptides had 10 or fewer amino acid residues, whereas the remaining 35 were longer but within 20 residues. Within the group of 51 stable peptides, 17 had a PeptideRanker score above 0.5 and were therefore considered bioactive candidates. These findings are consistent with previous in silico screenings of food-derived peptides, such as oat kernel peptides evaluated by Darewicz et al. [20] using a comparable computational workflow. The 17 stable bioactive peptides were subsequently docked against three targets relevant to hyperglycaemia: α-glucosidase, α-amylase, aldose reductase, and one target related to glycation, RAGE. Among these, two sequences, peptide 1 (MQIFVK), inhibited all three diabetes related targets; however, it did not inhibit RAGE. Whereas peptide 2 inhibited all three enzymes and also RAGE. From a structure–activity perspective, the predominance of hydrophobic, proline-rich and aromatic amino acid residues in these sequences likely contributes to their stability and binding specificity. Such residues support π–π stacking, van der Waals interactions and hydrogen bonding with catalytic and substrate-recognition residues in the target enzymes, which may account for their superior docking performance relative to several previously reported food-derived peptides. Compared with other plant-based milks, such as soy and oat, which mainly yield single-target carbohydrase inhibitors [39], the dual interaction of coconut peptides with aldose reductase and RAGE, in addition to α-glucosidase and α-amylase, highlights their potential to attenuate both hyperglycaemia and glycation-associated oxidative stress. The discussion is well conducted and well supported with valid and contemporary references as well as the conclusion is in line with the presented results. However, some important information for reproducing the study is missing and should be completed. In particular regarding material and procedures:
3.1. Materials: Please discriminate brand and origin of each the mentioned reagents. Also, please specify brand, model and local of origin of the different analytical machines used in the experiment.
Please make sure that in vitro, in vivo and in silico are italicized throughout the manuscript.
Please, make sure that the nomenclature of the tested peptides for enzyme inhibition is coherent throughout the manuscript: peptide 1 and peptide 2. Sometimes it is referred as peptide one or peptide two by extense.
Author Response
Reviewer 03
This work titled "Coconut milk-derived bioactive peptides as multifunctional agents against hyperglycemia, oxidative stress, and glycation: An integrated experimental and computational study" mostly shows an alternative research focusing on natural resources in the quest for a therapy that integrates anti-hyperglycemic, antioxidant, and anti-glycation properties, along with potent inhibition of α-glucosidase, α-amylase, and aldose reductase, as well as suppression of AGEs through RAGE, can be optimal in controlling diabetes and its complications. Authors, based on the literature, indicate that chemical analyses indicate that CM is rich in nutritionally and biologically active constituents, such as phenolic derivatives, a variety of fatty acids, antioxidant molecules, simple sugars, complex carbohydrates, and structural proteins [13]. From there, authors developed this work aiming to study the anti-hyperglycemic and anti-glycation properties of coconut milk bioactive peptides using in silico and in vitro approaches, which will aid future research in developing an alternative food-derived bioactive molecule to reduce glucose and, consequently, advanced glycation end products.
According to the authors, the protein analysis show results are in line with the previous study by Kwon et al. [16], which identified five different proteins: albumins, globulins, prolamines, glutelins-1, and glutelins-2.The predominant protein in coconut (65%) is 11S globulin, known as Cocosin [19]. Cocosin is thought to be more crucial in maintaining the stability of coconut milk compared to albumin or the 7S globulin fraction [18]. After protein extraction and SDS-PAGE, peptides were identified using nano-ESI-Orbitrap-LC-MS/MS, and their sequences were compared with the Cocos nucifera proteome library. A total of 114 peptide sequences have been identified, of which 96 unique peptides were retained after removal of duplicates - were predicted to be stable, with an instability index below 40. Seventeen of these stable peptides had 10 or fewer amino acid residues, whereas the remaining 35 were longer but within 20 residues. Within the group of 51 stable peptides, 17 had a PeptideRanker score above 0.5 and were therefore considered bioactive candidates. These findings are consistent with previous in silico screenings of food-derived peptides, such as oat kernel peptides evaluated by Darewicz et al. [20] using a comparable computational workflow. The 17 stable bioactive peptides were subsequently docked against three targets relevant to hyperglycaemia: α-glucosidase, α-amylase, aldose reductase, and one target related to glycation, RAGE. Among these, two sequences, peptide 1 (MQIFVK), inhibited all three diabetes related targets; however, it did not inhibit RAGE. Whereas peptide 2 inhibited all three enzymes and also RAGE. From a structure–activity perspective, the predominance of hydrophobic, proline-rich and aromatic amino acid residues in these sequences likely contributes to their stability and binding specificity. Such residues support π–π stacking, van der Waals interactions and hydrogen bonding with catalytic and substrate-recognition residues in the target enzymes, which may account for their superior docking performance relative to several previously reported food-derived peptides. Compared with other plant-based milks, such as soy and oat, which mainly yield single-target carbohydrase inhibitors [39], the dual interaction of coconut peptides with aldose reductase and RAGE, in addition to α-glucosidase and α-amylase, highlights their potential to attenuate both hyperglycaemia and glycation-associated oxidative stress. The discussion is well conducted and well supported with valid and contemporary references as well as the conclusion is in line with the presented results. However, some important information for reproducing the study is missing and should be completed. In particular regarding material and procedures:
3.1. Materials: Please discriminate brand and origin of each the mentioned reagents. Also, please specify brand, model and local of origin of the different analytical machines used in the experiment.
Author’s response: As per the reviewer’s suggestion, the brand and origin of each reagent have been mentioned in the materials section of the manuscript. Also, we have specified the brand, model and location of origin of the analytical machines used in this study.
Please make sure that in vitro, in vivo and in silico are italicized throughout the manuscript.
Author’s response: As per the reviewer’s suggestion, the words in vitro, in vivo and in silico are italicized throughout the manuscript.
Please, make sure that the nomenclature of the tested peptides for enzyme inhibition is coherent throughout the manuscript: peptide 1 and peptide 2. Sometimes it is referred as peptide one or peptide two by extense.
Authors response: As per the reviewer’s suggestion, the nomenclature of the tested peptides for enzyme inhibition is coherent as peptide 1 and peptide 2 throughout the manuscript.
Overall, we hope these revisions address the reviewers' concerns to the best of our knowledge.
Round 2
Reviewer 1 Report
Comments and Suggestions for Authors
I do not have further questions. I recommend acceptance of the manuscript.